# Bayesian Preference Elicitation for Personalized Prefactual Recommendation

## Abstract

A prefactual recommendation, also known as an algorithmic recourse, provides actionable guidance to an individual to overturn a machine learning prediction at a minimal cost of efforts. Existing methods impose an explicit assumption on the cost function, but in reality, different individuals may possess diverse and unique cost preferences. Failing to adapt the guidance to an individual's cost preference can lead to irrelevant and inefficient recommendations. To personalize the guidance to the individual cost, we propose a Bayesian preference elicitation framework that learns the cost function from the individual's feedback on a small number of pairwise comparisons. This framework relies on a sequential, mutual-information-maximization question-answering scheme to obtain a posterior distribution of an individual's cost weighting matrix. We then deploy this posterior to recommend a graph-based sequential guidance with minimal expected cost, leading the individual to achieve the desired algorithmic outcome. Numerical experiments on synthetic and real-world datasets demonstrate the power of our method in capturing the individual's preference and recommending personalized recourse.

## 1 Introduction

Algorithmic decision-making has become an integral part of various domains, ranging from finance (Turkson et al., 2016; Wang et al., 2020), healthcare (Fatima & Pasha, 2017; Yu et al., 2021) to online platforms and recommendation systems (Chen et al., 2019; Khanal et al., 2020). In consequential domains, a prediction from the algorithm may exert long-term effects on the life and future of the people impacted by the prediction. If an individual receives an unfavorable prediction from the algorithm, the individual should be given the reason behind the algorithmic decision and the necessary steps to obtain an alternative algorithmic outcome. For instance, suppose a candidate is applying for a graduate study and is predicted to be uncompetitive for admission; the candidate should be informed about the recommended actions to get accepted, possibly in the next admission cycle. To equip the individual with the power to overturn the algorithmic prediction, prefactual recommendation, also known as algorithmic recourse, has emerged to provide individuals with actionable guidance to rectify the undesired outcomes.

In the realm of generating recourse for machine learning model predictions, a diversity of methods has been explored. Utilizing the gradient information of the underlying model, this method generates counterfactual instances that closely resemble the given input, a technique pioneered by Wachter et al. (2017). Building on this, Ustun et al. (2019) proposed an integer programming formulation, tailored for linear classifiers, to identify actionable changes for a specific input instance. This concept of actionable change was further extended by Poyiadzi et al. (2020), who constructed an actionability graph using training data, delineating sequences of steps to generate counterfactual explanations. In a similar vein, Pawelczyk et al. (2020) integrated manifold learning principles to derive counterfactual instances, focusing on high-density regions of the data. The discourse then advanced with Karimi et al. (2020), who introduced two distinct approaches: one leveraging Bayesian model averaging for uncertainty, and the other employing a subpopulation-based interventional notion. This evolution culminated in the work of Karimi et al. (2021), which introduced a paradigm shift towards minimal interventions as a recourse, moving away from traditional nearest counterfactual explanations..

The aforementioned works typically rely on explicit assumptions about the cost function, disregarding the inherent diversity of individual cost preferences. This neglect to tailor the guidance to an

individual's cost preference can result in inefficient and cost-suboptimal recommendations. To address this issue, Yadav et al. (2021) proposed a new method to identify the recourse sets without the need of prior knowledge of their cost functions. They introduced the Expected Minimum Cost objective function, which is guided by two fundamental principles. The first principle asserts that providing a single low-cost solution from a set of options is sufficient to meet the user's needs. The second principle acknowledges the lack of complete knowledge about the user's true cost function and proposes approximating user satisfaction by sampling plausible cost functions. The method aims to achieve satisfactory recommendations tailored to the individual's preferences by optimizing the favorable cost outcomes within the generated recourse set.

In this work, we focus on Bayesian preference elicitation (BPE), which aims at handling uncertainty in cost parameters. BPE entails modeling this uncertainty using a prior distribution and refining it based on individual response, ultimately minimizing the ambiguity associated with these cost parameters (Viappiani & Boutilier, 2010). Xie et al. (2014) introduced a Bayesian uncertainty-based framework to learn individuals' preferences by incorporating a mixture of Gaussian prior beliefs. They utilized choice comparison queries to elicit user preferences and updated the posterior distribution using constrained sampling strategies based on the responses. Furthermore, various approaches have been proposed for selecting queries within BPE. One common criterion for question selection is expected value information (EVOI), as utilized in works such as Chajewska et al. (2000), Boutilier (2002), and Vendrov et al. (2020). EVOI aims to select the question that maximizes the expected utility function with respect to the posterior distribution (Howard, 1984). Another approach involves selecting questions based on the maximum mutual information, where the expected response provides the highest information gain (Canal et al., 2019; Rokach & Kisilevich, 2012). From a computational perspective, methods based on mutual information tend to be more tractable than EVOI-based approaches (Vendrov et al., 2020). Bayesian preference elicitation offers robust cost parameter predictions in the presence of noisy answers (Guo & Sanner, 2010; De Toni et al., 2022).

**Our Contributions:** In this paper, we introduce a Bayesian preference elicitation framework to capture the unique cost preferences of individuals. Leveraging the posterior distribution obtained from the elicitation process, we generate a graph-based sequential guidance for each individual that effectively attains a favorable algorithmic outcome at a low personal cost. We summarize our contributions below:

- We develop a probabilistic scheme for question selection that maximizes the mutual information between the user's response and the cost function parameter. Our method allows for efficient question selection as mutual information can be analytically calculated.
- We present a belief updating formulation considering both prior-posterior distortions and response alignment. After a proper compactification of the Bayesian posterior update problem, we develop a projected gradient descent algorithm to find the best parameters for the posterior.
- We propose a graph-based sequential guidance scheme that leverages the posterior distribution to minimize the expected recourse cost. The sequential guidance problem can be formulated as a binary program and solved by off-the-shelf solvers to a sufficiently large problem size.
- We conduct extensive numerical experiments on both synthetic and real-world datasets to demonstrate the efficiency of our Bayesian elicitation framework and graph-based recourse recommendation approach.

## 2 PROBLEM STATEMENT

We have a binary classifier $\mathcal{C} : \mathbb{R}^d \to \{0, 1\}$ and access to a training dataset consisting of $N + M$ data samples with features $x_i \in \mathbb{R}^d$, where $i = 1, \ldots, N + M$. The dataset is divided into two parts: a positive dataset $\mathcal{D}_1 = \{x_1, \ldots, x_N\}$ containing all instances for which $\mathcal{C}(x_i) = 1$ holds, and a negative dataset $\mathcal{D}_0 = \{x_{N+1}, \ldots, x_{N+M}\}$ containing instances with a negative predicted outcome, i.e., instances for which $\mathcal{C}(x_i) = 0$ holds. For a subject with input $x_0 \in \mathbb{R}^d$ receiving a negative prediction $\mathcal{C}(x_0) = 0$, we assume a Mahalanobis cost function $c_{A_0}(x, x_0) = (x - x_0)^\top A_0 (x - x_0)$, with $A_0$ being a positive semidefinite matrix.

Our framework employs this Mahalanobis distance to personalize the measure between the human representation $x_0$ and a specific choice $x_i$, capturing subjective differences by extending Euclidean distance with a linear transformation, akin to metric learning (Kulis, 2013; Yang & Jin, 2006; Bellet

et al., 2013). This minimization aligns with our goal of efficiently achieving preferred choices. When selecting $x_r$, our primary criteria are to ensure a positive projected outcome $\mathcal{C}(x_r) = 1$ and minimize the cost $c_{A_0}(x_r, x_0)$.

We propose a Bayesian preference elicitation framework to learn the cost function $c_{A_0}(x, x_0)$, where $A_0$ is an unknown matrix. This framework allows us to estimate preferences by incorporating prior knowledge and updating our beliefs based on observed responses. Before introducing the entire framework, we first review the definition of the Wishart distribution, which plays a crucial role in establishing our prior belief for the cost function. The selection of the Mahalanobis distance is rooted in the critical necessity for a weighting matrix to be positive definite, ensuring the preservation of meaningful geometric attributes (Gelman et al., 1995). We adopt the Wishart distribution, a prevalent choice in Bayesian modeling and machine learning to fulfill the necessity for positive definiteness, a characteristic foundational to its application (Dawid, 1981; Hurtado Rua et al., 2015) and widely used in diverse practical contexts.

**Definition 2.1** (Wishart distribution). *A probability measure $\mathbb{P}$ on the space of $d$-by-$d$ symmetric positive semidefinite matrices $\mathbb{S}_+^d$ is said to be a Wishart measure with degrees of freedom $m \in \mathbb{N}_+$, $m \geq d$ and scale matrix $\Sigma \in \mathbb{S}_{++}^d$ if it admits a density function*

$$f_{\mathbb{P}}(A) = \frac{1}{2^{\frac{md}{2}} \det(\Sigma)^{\frac{m}{2}} \Gamma_d(\frac{m}{2})} \det(A)^{\frac{m-d-1}{2}} \exp(-\frac{1}{2}\langle \Sigma^{-1}, A\rangle),$$

*where $\Gamma_d$ is the $d$-variate Gamma function computed as $\Gamma_d\left(\frac{n}{2}\right) = \pi^{d(d-1)/4} \prod_{j=1}^d \Gamma\left(\frac{n}{2} - \frac{j-1}{2}\right).$*

We write $\mathbb{P} \sim \mathcal{W}_d(m, \Sigma)$ to denote that $\mathbb{P}$ is a Wishart measure. While the general definition of Wishart distributions allows for $m$ to be a real number, for simplicity, our work restricts $m$ to an integer.

Our approach begins with Bayesian preference elicitation, where we represent our prior belief about the random matrix $\tilde{A}$ using a Wishart distribution $\mathcal{W}_d(m, \Sigma)$. Over $T$ rounds of cost elicitation, we strategically select pairs from the positive dataset $\mathcal{D}_1$ to gather subject preferences based on the cost function. We optimize question pairs to maximize mutual information between responses and $\tilde{A}$, leveraging our prior knowledge. This informs our posterior inference, allowing us to refine our belief about $\tilde{A}$. Finally, upon completing the $T$ rounds, we generate recourse recommendations based on the final posterior distribution of $\tilde{A}$.

In our notation, $\mathbb{S}^d$, $\mathbb{S}_+^d$, and $\mathbb{S}_{++}^d$ denote spaces of $d$-by-$d$ symmetric matrices, positive semi-definite matrices, and positive definite matrices, respectively. We use $I$ for the identity matrix. For any $X$ in $\mathbb{R}^{d \times d}$, we define several operations: the trace operator as $\text{Tr}[X] = \sum_{i=1}^d X_{ii}$, the Frobenius norm as $\|X\|_F$, the determinant as $\det(X)$, and the vectorization of $X$ as $\text{vec}(X)$. The inner product between matrices $X$ and $Y$ in $\mathbb{S}^d$ is represented as $\langle X, Y \rangle = \text{Tr}[XY]$. The notation $X \succeq Y$ indicates $X - Y \in \mathbb{S}_+^d$. We use $\otimes$ for the outer product and $\mathbb{I}_C(\cdot)$ to denote the indicator function of set $C$. In terms of specific matrices relevant to our study, we employ $A_0$ to signify a deterministic matrix specific to the user, while $\tilde{A}$ is a random matrix representing our belief about the user's actual matrix $A_0$. Subsequently, $A$ corresponds to a realization of $\tilde{A}$.

## 3 QUESTION DETERMINATION

In our work, we primarily focus on pairwise comparisons, with an extension to listwise comparisons in Appendix A, which entail higher computational complexity in our framework. Pairwise comparisons offer advantages, including reduced user cognitive load (Payne et al., 1993) and increased robustness to noisy responses and uncertain preferences (Plackett, 1975), aligning well with our approach. At each time step $t$, our objective is to identify the optimal pair of questions, denoted as $x_i$ and $x_j$, to query. To achieve this, we make the assumption that the response obtained from questions $x_i$ and $x_j$ adheres to the Bradley-Terry-Luce (BTL) model (Bradley & Terry, 1952): Given a pair of questions $(x_i, x_j)$ and $A \in \mathbb{S}_+^d$, define the cost difference $\Delta_{ij}(A) \triangleq c_A(x_i, x_0) - c_A(x_j, x_0)$. We posit that the user response $\tilde{R}_{ij}^\kappa(A)$ can be represented by the probabilistic model:

$$\tilde{R}_{ij}^\kappa(A) = \begin{cases} +1 & \text{with probability } 1 - \Phi(\kappa \Delta_{ij}(A)), \\ -1 & \text{with probability } \Phi(\kappa \Delta_{ij}(A)), \end{cases} \tag{1}$$

where $\kappa > 0$ is the slope parameter and $\Phi(v) = 1/(1 + e^{-v})$ is the link function. The BTL model has applications in various machine learning domains, including reinforcement learning from human feedback (RLHF) (Christiano et al., 2017; Ouyang et al., 2022), recommendation system rank aggregation (Agarwal et al., 2020), and ordinal embedding tasks (Ghosh et al., 2019). The slope parameter $\kappa$ controls the model's sensitivity to differences in cost between inputs. A higher $\kappa$ increases the sensitivity, while a lower $\kappa$ reduces it, implying a higher degree of noise in the response. As $\kappa$ approaches infinity, the response relies solely on input cost differences.

We aim to maximize the mutual information between the response $\tilde{R}_{ij}^{\kappa}$ and the weighting matrix $\tilde{A}$ to gain the most information about $\tilde{A}$ through observations. Let $H(\cdot)$ denote the entropy of a random variable; the mutual information between $\tilde{A}$ and $\tilde{R}_{ij}^{\kappa}$ is defined as:

$$MI(\tilde{A}, \tilde{R}_{ij}^{\kappa}) = H(\tilde{R}_{ij}^{\kappa}) - H(\tilde{R}_{ij}^{\kappa}|\tilde{A}).$$

However, computing the entropy is challenging due to the continuous density of $\tilde{A}$. Existing methods rely on sampling (Gao et al., 2017; Mesner & Shalizi, 2020): they involve sampling the matrices from the positive semidefinite space and the subsequent responses based on the model (1). This approach results in a heavy computational burden as it requires multiple iterations to accumulate sufficient samples. The following section explores the asymptotic mutual information, which admits an analytical expression. This analytical expression presents computational advantages and eliminates the need for extensive sampling.

## 3.1 COMPUTING THE MUTUAL INFORMATION

To efficiently compute mutual information, we consider the limit as $\kappa$ approaches infinity, which corresponds to the noiseless case of the BTL model. Before stating the following proposition, we introduce the term $\gamma_{ij}^{\kappa} \triangleq \mathrm{Prob}(\tilde{R}_{ij}^{\kappa}(\tilde{A}) = +1)$. This probability accounts for two layers of randomness: one arising from the matrix $\tilde{A}$, and the other stemming from the $\tilde{R}_{ij}^{\kappa}$ following the BTL model after the realization of $\tilde{A}$.

**Proposition 3.1** (Asymptotic mutual information). *Let $\mathbb{P}_{t-1}$ represent the prior distribution of $\tilde{A}$ at time $t$ and assuming $\mathbb{P}_{t-1}$ has a continuous density, and define $\gamma_{ij} \triangleq \lim_{\kappa \to \infty} \gamma_{ij}^{\kappa}$. Then $\gamma_{ij} = \mathbb{P}_{t-1}(\Delta_{ij}(\tilde{A}) \leq 0)$ and*

$$\lim_{\kappa \to +\infty} MI(\tilde{A}, \tilde{R}_{ij}^{\kappa}) = -\gamma_{ij} \log_2(\gamma_{ij}) - (1 - \gamma_{ij}) \log_2(1 - \gamma_{ij}).$$

To evaluate $\gamma_{ij}$, note that $\Delta_{ij}(\tilde{A}) = (x_i - x_0)^\top \tilde{A}(x_i - x_0) - (x_j - x_0)^\top \tilde{A}(x_j - x_0)$. Recall that the belief of $\tilde{A}$ follows a Wishart distribution, and by the property of Wishart distribution (Rao, 2009, Section 8b.2), we have that

$$(x_i - x_0)^\top \tilde{A}(x_i - x_0) \sim \sigma_i \chi_m^2, \quad (x_j - x_0)^\top \tilde{A}(x_j - x_0) \sim \sigma_j \chi_m^2,$$

where $\sigma_i^2 = (x_i - x_0)^\top \Sigma(x_i - x_0)$, $\sigma_j^2 = (x_j - x_0)^\top \Sigma(x_j - x_0)$, and $\chi_m^2$ is a chi-squared distribution with $m$ degrees of freedom. Then $\Delta_{ij}(\tilde{A})$ is the difference of two gamma random variables. We establish the following theorem for computing $\mathbb{P}_{t-1}(\Delta_{ij}(\tilde{A}) \leq 0)$ with this property.

**Theorem 3.2** (Probability value). *Suppose that $\mathbb{P}_{t-1} \sim \mathcal{W}_d(m, \Sigma)$. For any $x_i$ and $x_j$ such that $z_i \triangleq x_i - x_0$ and $z_j \triangleq x_j - x_0$ are not parallel to each other, define the following quantities:*

$$\sigma_i^2 \triangleq z_i^\top \Sigma z_i, \ \sigma_j^2 \triangleq z_j^\top \Sigma z_j, \quad and \quad \rho \triangleq \frac{[z_i \otimes z_i]^\top [\Sigma \otimes \Sigma][I_{d^2} + C][z_j \otimes z_j]}{2\sigma_i^2 \sigma_j^2},$$

*where $C \in \mathbb{R}^{d^2 \times d^2}$ is a commutation matrix, i.e., $C = \sum_{i=1}^{d} \sum_{j=1}^{d} [e_j \otimes e_i][e_i \otimes e_j]^\top$, with $e_i$ denotes the $i$-th column vector of the identity matrix $I_d$. Then $\rho \in [0, 1)$ and*

$$\mathbb{P}_{t-1}(\Delta_{ij}(\tilde{A}) \leq 0) = (\frac{1+c}{1-c})^{a+\frac{1}{2}} \frac{\Gamma(2a+1)\Gamma(1)}{\Gamma(a+\frac{3}{2})\Gamma(a+\frac{1}{2})} \, {}_2F_1\Big(2a+1, a+\frac{1}{2}; a+\frac{3}{2}; -\frac{1+c}{1-c}\Big), \quad (2)$$

*where the parameters $a$, $b$, and $c$ admit values*

$$a = \frac{m-1}{2}, \quad b = \frac{8\sigma_i^2\sigma_j^2(1-\rho)}{\sqrt{4(\sigma_i^2 - \sigma_j^2)^2 + 16\sigma_i^2\sigma_j^2(1-\rho)}}, \quad c = -\frac{2(\sigma_i^2 - \sigma_j^2)}{\sqrt{4(\sigma_i^2 - \sigma_j^2)^2 + 16\sigma_i^2\sigma_j^2(1-\rho)}},$$

*and ${}_2F_1$ is the Gauss' hypergeometric function.*

Theorem 3.2 provides an analytical expression for computing $\mathbb{P}_{t-1}(\Delta_{ij}(\tilde{A}) \leq 0)$, offering significant computational advantages because we do not need to run time-consuming, intensive simulations to estimate this quantity. In fact, for each sample $\hat{A}$ drawn from a Wishart distribution, the computational complexity of evaluating $(x_i - x_0)^\top \hat{A}(x_i - x_0)$ is $\mathcal{O}(d^2)$, resulting in an overall complexity of $\mathcal{O}(Ld^2)$ for an empirical estimator with $L$ samples. In contrast, the analytical expression derived in Theorem 3.2 has a computational complexity similar to that of computing $\rho$. The calculation of the quantity $\rho$ requires performing Kronecker product operations. Specifically, we obtain intermediate results by calculating $[z_i \otimes z_i]^\top [\Sigma \otimes \Sigma]$ and $[I_{d^2} + C][z_j \otimes z_j]$. Notably, $[z_i \otimes z_i]^\top [\Sigma \otimes \Sigma] = [\Sigma z_i \otimes \Sigma z_i]^\top$, allowing us to avoid the direct computation of $\Sigma \otimes \Sigma$. This approach results in a computational complexity of $\mathcal{O}(d^2)$ for the analytical expression, significantly lower than the complexity of the sampling method.

## 3.2 QUESTION SELECTION

To select the next question, we aim to identify a pair of indices $(i, j) \in [N] \times [N]$, representing two inputs $x_i$ and $x_j$, drawn from the positive dataset $\mathcal{D}_1$. Our primary goal is to maximize the mutual information associated with the question pair $(x_i, x_j)$. While $\kappa$ is finite in practice, our framework selects the pair $(x_i, x_j)$ based on the asymptotic mutual information with $\kappa = +\infty$. Empirical results in Appendix D.5 demonstrate that our framework performs effectively with finite $\kappa$, indicating convergence of the posterior distribution to the ground truth $A_0$ in such cases. The strategy for selecting the question $(x_i, x_j)$ involves a nested iteration over the positive dataset $\mathcal{D}_1$. This approach exhibits a computational complexity of $\mathcal{O}(N^2)$, where $N$ represents the cardinality of $\mathcal{D}_1$.

## 4 POSTERIOR UPDATE

This section focuses on estimating the posterior distribution $\mathbb{P}_t$ given the prior distribution $\mathbb{P}_{t-1}$ at time $t$ and the response $R_{ij} \in \{+1, -1\}$. We aim to update the distribution based on the available information and refine our belief of the cost matrix $A_0$.

### 4.1 PROBLEM FORMULATION OF POSTERIOR UPDATE

Before formulating the posterior update problem, we revisit the Kullback-Leibler (KL) divergence.

**Definition 4.1** (Kullback-Leibler (KL) divergence). *Given two distributions $\mathbb{P}$ and $\mathbb{Q}$ such that $\mathbb{P}$ is absolutely continuous with respect to $\mathbb{Q}$, the KL divergence from $\mathbb{P}$ to $\mathbb{Q}$ is $\mathrm{KL}(\mathbb{P} \| \mathbb{Q}) \triangleq \mathbb{E}_{\mathbb{P}}[\log d\mathbb{P}/d\mathbb{Q}]$, where $d\mathbb{P}/d\mathbb{Q}$ is the Radon-Nikodym derivative of $\mathbb{P}$ with respect to $\mathbb{Q}$.*

Let $M_{ij} = (x_i - x_0)(x_i - x_0)^\top - (x_j - x_0)(x_j - x_0)^\top$. It is evident that $\langle M_{ij}, \tilde{A} \rangle = \Delta_{ij}(\tilde{A})$. We propose to use an approximate posterior that is determined by solving:

$$
\begin{aligned}
\min \quad & \mathrm{KL}(\mathbb{P} \| \mathbb{P}_{t-1}) + \tau \kappa \mathbb{E}_{\mathbb{P}}[R_{ij} \langle M_{ij}, \tilde{A} \rangle] \\
\text{s.t.} \quad & \mathbb{P} \sim \mathcal{W}_d(m, \Sigma), \ m \in \mathbb{N}_+, \ \Sigma \in \mathbb{S}_{++}^d \\
& \mathrm{Tr}[\Sigma] = d, \ d \leq m \leq m_{t-1}.
\end{aligned}
\tag{3}
$$

Two primary objectives guide our formulation. First, we seek to minimize the distortion between the prior distribution $\mathbb{P}_{t-1}$ and the posterior distribution $\mathbb{P}_t$ by quantifying the KL divergence. This approach draws inspiration from stochastic variational inference techniques (Hoffman et al., 2013; Bottou, 2010), where the KL divergence measures the proximity between the variational distribution and the posterior distribution. We choose the reverse KL divergence $\mathrm{KL}(\mathbb{P} \| \mathbb{P}_{t-1})$ over the forward KL divergence $\mathrm{KL}(\mathbb{P}_{t-1} \| \mathbb{P})$ based on Bayes' theorem (Schervish, 2012, Theorem 1.31). This theorem establishes that the posterior distribution is absolutely continuous with respect to the prior distribution, ensuring that the reverse KL divergence is well-defined.

Secondly, our main goal is to maximize the likelihood $\mathrm{Prob}(\tilde{R}_{ij}^\kappa(\tilde{A}) = R_{ij})$ to align the observed response with preferences represented by $\tilde{A}$. This is equivalent to minimizing the probability $\mathrm{Prob}(\tilde{R}_{ij}^\kappa(\tilde{A}) \neq R_{ij})$, which can be expressed as:

$$
\mathrm{Prob}(\tilde{R}_{ij}^\kappa(\tilde{A}) \neq R_{ij}) = \int_{\mathbb{S}_+^d} (1 - \Phi(-\kappa R_{ij} \Delta_{ij}(S))) f_{\mathbb{P}}(S) \mathrm{d}S = \int_{\mathbb{S}_+^d} \Phi(\kappa R_{ij} \Delta_{ij}(S)) f_{\mathbb{P}}(S) \mathrm{d}S.
$$

The last equality follows from the property $\Phi(-v) = 1 - \Phi(v)$. The above integral is computationally intensive to compute, so we approximate the function $\Phi(v)$ by a linear function $v \mapsto v$, resulting in an approximation:

$$\text{Prob}(\tilde{R}_{ij}^{\kappa}(\tilde{A}) \neq R_{ij}) \approx \int_{\mathbb{S}_+^d} \kappa R_{ij} \Delta_{ij}(S) f_{\mathbb{P}}(S) \mathrm{d}S = \kappa \mathbb{E}_{\mathbb{P}}[R_{ij} \Delta_{ij}(\tilde{A})] = \kappa \mathbb{E}_{\mathbb{P}}[R_{ij} \langle M_{ij}, \tilde{A} \rangle].$$

To balance fidelity to the prior distribution with alignment with observed responses, we introduce the positive parameter $\tau$. Finally, problem (3) imposes a Wishart parametric form on the posterior to facilitate successive updating steps. We leverage the following fact to reformulate the problem (3) into one that exclusively involves the parameters $m$ and $\Sigma$.

**Fact 4.2** (KL divergence between Wishart distributions (Penny, 2001)). *Let $\mathbb{P} \sim \mathcal{W}_d(m_p, \Sigma_p)$ and $\mathbb{Q} \sim \mathcal{W}_d(m_q, \Sigma_q)$ be two Wishart distributions $\mathbb{S}_+^d$. The KL divergence from $\mathbb{P}$ to $\mathbb{Q}$ amounts to*

$$\text{KL}(\mathbb{P} \,\|\, \mathbb{Q}) = -\frac{m_q}{2} \log \det \left( \Sigma_q^{-1} \Sigma_p \right) + \frac{m_p}{2} \left( \text{Tr}[\Sigma_q^{-1} \Sigma_p] - d \right) + \log \frac{\Gamma_d \left( \frac{m_q}{2} \right)}{\Gamma_d \left( \frac{m_p}{2} \right)} + \frac{m_p - m_q}{2} \psi_d \left( \frac{m_p}{2} \right),$$

*where $\psi_d$ is the multivariate digamma function.*

When the prior distribution is a Wishart distribution $\mathbb{P}_{t-1} \sim \mathcal{W}_d(m_{t-1}, \Sigma_{t-1})$, Fact 4.2 asserts that the loss function of problem (3) is

$$
\begin{aligned}
L(m, \Sigma) = &- \frac{m_{t-1}}{2} \log \det \left( \Sigma_{t-1}^{-1} \Sigma \right) + \frac{m}{2} \left( \text{Tr}[\Sigma_{t-1}^{-1} \Sigma] - d \right) + \log \frac{\Gamma_d \left( \frac{m_{t-1}}{2} \right)}{\Gamma_d \left( \frac{m}{2} \right)} \\
&+ \frac{m - m_{t-1}}{2} \psi_d \left( \frac{m}{2} \right) + \tau \kappa m \, \text{Tr}[R_{ij} M_{ij} \Sigma].
\end{aligned}
$$

If we fix a value of the integer $m$ with $d \leq m \leq m_{t-1}$ and solve only over the matrix variable $\Sigma$, then we obtain the equivalent optimization problem

$$
\begin{aligned}
\min \quad & \ell(\Sigma) \triangleq -\frac{m_{t-1}}{m} \log \det(\Sigma) + \text{Tr}[(\Sigma_{t-1}^{-1} + \tau \kappa R_{ij} M_{ij}) \Sigma] \\
\text{s.\,t.} \quad & \Sigma \in \mathbb{S}_{++}^d, \ \text{Tr}[\Sigma] = d.
\end{aligned}
\tag{4}
$$

Problem (4) has a non-empty feasible set: in fact, the identity matrix is a feasible solution. Nevertheless, the feasible set of (4) is open due to the constraint $\Sigma \in \mathbb{S}_{++}^d$. The next result asserts a compactification of this feasible set.

**Proposition 4.3** (Compactification). *Problem (4) is equivalent to*

$$\min \left\{ \ell(\Sigma) \ : \ \Sigma \in \mathbb{S}_+^d, \ \text{Tr}[\Sigma] = d, \ \Sigma \succeq \varepsilon I \right\}, \tag{5}$$

*where the constant $\varepsilon$ is $\varepsilon \triangleq d^{1-d} \exp(-\frac{m}{m_{t-1}} \| \Sigma_{t-1}^{-1} + \tau \kappa R_{ij} M_{ij} \|_F (\sqrt{d} + d)) > 0$.*

Problem (5) has a compact feasible set and is amenable to a projected gradient descent algorithm. Let $\Sigma^*(m)$ be the solution to (5) for a fixed $m$. Our problem remains to infer the optimal integer $d \leq m^* \leq m_{t-1}$ such that $(m^*, \Sigma^*(m^*))$ minimizes the loss function $L(m, \Sigma)$. It is achieved by solving problem (5) for all admissible values of $m$.

## 4.2 PROJECTED GRADIENT DESCENT ALGORITHM FOR POSTERIOR UPDATE

We apply the projected gradient method to solve problem (5). Let $D$ be the feasible set of problem (5), defined as $D \triangleq \{\Sigma \in \mathbb{R}^{d \times d} : \Sigma \in \mathbb{S}_+^d, \ \text{Tr}[\Sigma] = d, \ \Sigma \succeq \varepsilon I\}$. We now study the projection onto the feasible set of (5). For any symmetric matrix $S$, define the projection operator on the feasible set

$$\text{Proj}(S) \triangleq \min\{\|S - \Sigma\|_F^2 : \Sigma \in \mathbb{S}_+^d, \ \Sigma \succeq \varepsilon I, \ \text{Tr}[\Sigma] = d\}.$$

The following lemma demonstrates that the projection onto the set $D$ can be simplified to the projection onto a simplex in $\mathbb{R}^d$.

**Lemma 4.4** (Projection operator). *For any symmetric matrix $S$ with the eigendecomposition $S = V \text{diag}(s) V^\top$, where $s \in \mathbb{R}^d$ is the vector of eigenvalues, we have $\text{Proj}(S) = V \text{diag}(\lambda^\star + \varepsilon \mathbf{1}) V^\top$, where $\mathbf{1}$ is a $d$-dimensional vector of all ones and $\lambda^\star$ solves the projection onto the simplex problem*

$$\lambda^\star = \arg\min \left\{ \| \lambda - (s - \varepsilon \mathbf{1}) \|_2^2 : \lambda \geq 0, \ \lambda^\top \mathbf{1} = (1 - \varepsilon)d \right\}. \tag{6}$$

The steps of the projected gradient descent algorithm are summarized in Algorithm 1. For brevity, we omit the details of projection onto $D$. We refer the reader to the Appendix C.4 for further details.

We prove the strong convexity and Lipschitz continuity of the gradient of the objective function $\ell$, which are crucial for the convergence guarantee of Algorithm 1.

**Lemma 4.5** (Smoothness). *The objective function $\ell$ is strongly convex with parameter $m_{t-1}/(md^2)$ and has a Lipschitz continuous gradient with Lipschitz constant $m_{t-1}/(m\varepsilon^2)$ on $D$.*

By Lemma 4.5, Algorithm 1 exhibits linear convergence rate (Beck, 2017, Theorem 10.29).

---

**Algorithm 1** Projected gradient descent to solve (5)

---

**Input:** Prior scaling matrix $\Sigma_{t-1} \in \mathbb{S}_{++}^d$, degrees of freedom $m$
**Parameter:** Constant learning rate $t > 0$, Number of iterations: $K \in \mathbb{N}_+$
**Initialization:** Set $\Sigma_{(0)} \leftarrow \Sigma_{t-1}$
    **for** $k = 0, \ldots, K - 1$ **do**
        Compute gradient:
        $g \leftarrow -\frac{m_{t-1}}{m}\Sigma_{(k)}^{-1} + \Sigma_{t-1}^{-1} + \tau\kappa R_{ij}M_{ij}$
        Gradient descent and projection:
        $\Sigma_{(k+1)} \leftarrow \mathrm{Proj}(\Sigma_{(k)} - tg)$
    **end for**
**Output:** $\Sigma_{(K)}$

---

## 5 RECOURSE RECOMMENDATION

This section presents a model-agnostic recourse-generation method inspired by FACE (Poyiadzi et al., 2020). The central concept of this method involves two key steps. First, we construct a graph representation based on the given dataset, where each data point corresponds to a node in the graph. Second, we employ a shortest path algorithm to find the optimal path from the initial input $x_0$ to a node that yields a positive prediction according to the underlying machine learning model. Following this path, we generate a recourse that suggests a feasible and actionable counterfactual explanation.

The strategy for constructing the graph is as follows. We begin by creating a directed graph $\mathcal{G} = (\mathcal{V}, \mathcal{E})$, which captures the underlying geometric structure of the available data. In this graph, each node $x_i \in \mathcal{V}$ corresponds to a sample from the training set, while an edge $(x_i, x_j) \in \mathcal{E}$ represents a feasible transition from node $x_i$ to node $x_j$. By constructing this graph, we establish the connectivity between different samples and enable the exploration of potential paths for sequential recourse.

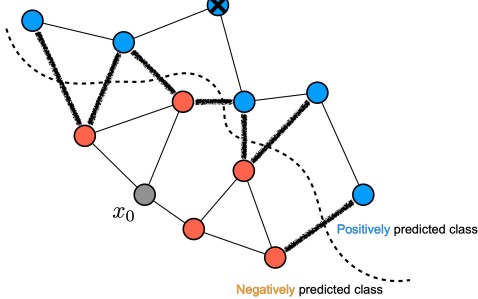

Figure 1: Visualization of a graph $\mathcal{G}$, decision boundary, recourse search. Problem (7a) seeks a path from $x_0$ (grey) to a positive prediction (blue). Crossed blue nodes with no links to negatives can be pruned.

Recall that $x_0$ is classified by $\mathcal{C}$ as negative: $\mathcal{C}(x_0) = 0$. After $T$ rounds of cost-elicitation, we obtain a Wishart posterior distribution $\mathbb{P}_T \sim \mathcal{W}_d(m_T, \Sigma_T)$ representing our belief about the subject's cost matrix $A_0$. Suppose we are given a graph $\mathcal{G} = (V, E)$, where $V$ is the node set and $E$ is the edge set. For each edge $e$, we use $o(e)$ and $d(e)$ to represent the origin and destination of this edge. The cost on edge $e$ is

$$c_e = (x_{o(e)} - x_{d(e)})^\top A(x_{o(e)} - x_{d(e)}) = \langle A, M_e \rangle,$$

where $M_e = (x_{o(e)} - x_{d(e)})(x_{o(e)} - x_{d(e)})^\top$. Any path from $x_0$ to a positively-predicted sample can be represented by a binary vector $z \in \{0, 1\}^{|E|}$. The cost of this path is a random variable $\langle A, \sum_{e \in E} M_e z_e \rangle$, where the source of randomness is in $A$. Supposing risk neutrality, we propose to find the recourse by solving the stochastic optimization problem

$$\min_{z \in \mathcal{Z}} \mathbb{E}_{\mathbb{P}_T}\left[\left\langle \sum_{e \in E} M_e z_e, A \right\rangle\right], \tag{7a}$$

that finds the sequential recourse with minimum expected cost. The set $\mathcal{Z}$ contains all possible paths that (i) start from $x_0$, (ii) have an end-node in the positively-predicted class

$$\mathcal{Z} = \left\{ z \in \{0, 1\}^{|E|} : \begin{array}{l} \sum_{e \in E : o(e) = x_0} z_e = 1 \\ \sum_{e \in E : o(e) = x_i} z_e - \sum_{e \in E : d(e) = x_i} z_e = 0 \quad \forall x_i : \mathcal{C}(x_i) = 0, x_i \neq x_0 \\ \sum_{x_i \in V : \mathcal{C}(x_i) = 1} \sum_{e \in E : d(e) = x_i} z_e = 1 \end{array} \right\}.$$

The first constraint indicates that the sum of the outflows from $x_0$ is one. The second constraint is a flow conservation constraint, indicating that for each node other than $x_0$ that is predicted to be in the negative class (i.e., orange nodes in Figure 1), then the sum of the inflows equals the sum of the outflows. Thanks to the moment condition $\mathbb{E}_{\mathbb{P}_T}[A] = m_T \Sigma_T$ of the Wishart distribution and the linearity of the expectation operator, the recourse finding problem above is equivalent to

$$\min_{z \in \mathcal{Z}} \left\langle \sum_{e \in E} M_e z_e, \Sigma_T \right\rangle, \tag{7b}$$

where we have dropped the parameter $m_T$ from the objective function because $m_T > 0$. Problem (7b) is a binary linear program that can be solved effectively using solvers like Mosek and GUROBI.

## 6 NUMERICAL EXPERIMENTS

We conducted extensive numerical experiments on synthetic and real-world datasets to evaluate our method: Prefactual Recommendation by Bayesian Preference Elicitation (Bayesian PR). Specifically, we employed three real-world benchmark datasets: the *German*, *Bank*, *Student* datasets, which are commonly used in recourse methods research. Additionally, we conducted comparisons with the graph-based recourse recommendation method Feasible and Actionable Counterfactual Explanation (FACE) (Poyiadzi et al., 2020) and the non-graph-based recourse recommendation methods Wachter (Wachter et al., 2017) and DiCE (Mothilal et al., 2020).

### 6.1 EXPERIMENTAL SETTINGS

We provide an overview of our data processing procedures, ground truth matrix $A_0$ generation, and details regarding the architecture of the classifier $\mathcal{C}$. All experiments were conducted on a 2x20 core Xeon Gold 6248 2.50GHz. We utilized min-max normalization for continuous features and one-hot encoding for categorical features, following the approach outlined in Mothilal et al. (2020). To generate the ground truth matrix cost $A_0$ for each user $x_0$, we created a random $d \times d$ matrix $A$ with independent zero-mean, unit-variance normal distribution elements, and computed $A_0 = AA^\top$. For classification, we employed a three-layer MLP with 50 neurons in the first layer, followed by two hidden layers, each containing 20 neurons, and an output neuron with sigmoid activation. The dataset was split into an 80% training set and a 20% testing set. For FACE, Wachter, and DiCE, we adhere to the setups outlined in their respective papers. In our method, we set the hyperparameters $\tau$ and $\kappa$ such that $\tau\kappa = 1$.

### 6.2 GRAPH-BASED COST-ADAPTIVE RECOURSE

In this experiment, we utilize a graph-based recourse recommendation method from Section 5 to generate a path and evaluate it using path cost and validity metrics. For the path $P$ from $x_0$ to $x_r$, we consider two types of true costs on edges: Mahalanobis distance and $\ell_1$ norm. The path cost for Mahalanobis distance is defined as the sum of Mahalanobis distances of the edges in $P$ with respect to the ground truth $A_0$, expressed as $c_{A_0}(P) = \sum_{t=0}^{T-1} (x_{t+1} - x_t)^\top A_0 (x_{t+1} - x_t)$. For $\ell_1$ norm, the path cost is defined as the sum of $\ell_1$ norms of the edges in $P$, expressed as $c_{\ell_1}(P) = \sum_{t=0}^{T-1} \|x_{t+1} - x_t\|_1$. The recommended recourse $x_r$ (the terminal node in the recommended path) is valid if the classifier predicts it as positive, i.e., $\mathcal{C}(x_r) = 1$. The validity is the ratio of $x_r$ with positive prediction over the total number of $x_r$.

In Table 1, we represent the true cost through the Mahalanobis distance, consistent with the foundational principles of our framework. In this setup, our framework accurately specifies the cost function type to align with the Mahalanobis distance, whereas FACE misspecifies it. As a result, our method shows enhanced cost efficiency when compared to FACE, as evidenced by our empirical results. This improvement suggests that our approach identifies more direct paths from the initial point $x_0$ to the recommended recourse $x_r$, resulting in more efficient and practical recommendations.

On the other hand, in our analysis presented in Table 2, we consider scenarios where the true cost is described by the $\ell_1$ norm, introducing a case of misspecification in our framework's cost function. Notably, in these instances, FACE accurately specifies the cost function, aligning perfectly with the true underlying cost structure. Despite this advantage for FACE, our method exhibits comparable

Table 1: Comparison of cost and validity of Bayesian PR and FACE using Mahalanobis distance as the true cost.

| Method | Synthetic | | German | | Bank | | Student | |
|---|---|---|---|---|---|---|---|---|
| | Cost | Validity | Cost | Validity | Cost | Validity | Cost | Validity |
| FACE | $0.18 \pm 0.18$ | $1.00 \pm 0.00$ | $0.19 \pm 0.15$ | $1.00 \pm 0.00$ | $1.27 \pm 0.70$ | $1.00 \pm 0.00$ | $1.14 \pm 0.63$ | $1.00 \pm 0.00$ |
| Bayesian PR | $\mathbf{0.03} \pm 0.01$ | $1.00 \pm 0.00$ | $\mathbf{0.16} \pm 0.13$ | $1.00 \pm 0.00$ | $\mathbf{0.97} \pm 0.90$ | $1.00 \pm 0.00$ | $\mathbf{1.11} \pm 0.62$ | $1.00 \pm 0.00$ |

performance. It particularly stands out in the Synthetic and Student datasets, as detailed in Table 2. The superior results in these specific datasets underscore the robustness and adaptability of our approach, demonstrating its effectiveness in providing accurate recommendations even when there is a discrepancy between the presumed and actual cost functions. Additionally, a more extensive comparison with non-graph-based methods is presented in Tables 5 and 6 in Appendix D. These comparisons reveal that our method outperforms the non-graph-based approach in real-world datasets, further validating its practical utility and effectiveness in diverse scenarios.

Table 2: Comparison of cost and validity of Bayesian PR and FACE using $\ell_1$ norm as the true cost.

| Method | Synthetic | | German | | Bank | | Student | |
|---|---|---|---|---|---|---|---|---|
| | Cost | Validity | Cost | Validity | Cost | Validity | Cost | Validity |
| FACE | $0.36 \pm 0.12$ | $1.00 \pm 0.00$ | $\mathbf{0.45} \pm 0.19$ | $1.00 \pm 0.00$ | $\mathbf{1.05} \pm 0.35$ | $1.00 \pm 0.00$ | $1.20 \pm 0.35$ | $1.00 \pm 0.00$ |
| Bayesian PR | $\mathbf{0.24} \pm 0.09$ | $1.00 \pm 0.00$ | $0.45 \pm 0.20$ | $1.00 \pm 0.00$ | $1.13 \pm 0.17$ | $1.00 \pm 0.00$ | $\mathbf{1.20} \pm 0.18$ | $1.00 \pm 0.00$ |

### 6.3 MEAN RANK PERFORMANCE

In this section, we rank the recourses based on individual preferences using the mean rank metric (Bertsimas & O'Hair, 2013). We begin by ranking all recourses in the positive dataset $\mathcal{D}_1$ (with size $N$) according to their Mahalanobis distances from $x_0$ with respect to $A_0$. The recourse with the smallest distance receives rank 1, while the one with the largest distance is ranked $N$. Subsequently, we replace the ground truth $A_0$ with the mean $m_T \Sigma_T$ and rank the recourses in $\mathcal{D}_1$ again. We fetch the top-$K$ recourses and obtain the corresponding ranking w.r.t. $A_0$. For top-$K$ recourses, we write the mean rank $\bar{r}$ as $\bar{r} = (\sum_{i=1}^{K} r_i - r_{\min})/r_{\max}$ with $r_{\min} = \sum_{i=1}^{K} i = (K+1)K/2$ and $r_{\max} = \sum_{i=N-K+1}^{N} i = (2N - K + 1)$.

We generate plots in Figure 2 to display the mean rank across various question counts ($T \in [1, 10]$) with a fixed $K$ value of 30. The trend across all datasets generally exhibits a decrease in mean rank as the number of questions posed to $x_0$ increases. This behavior indicates that the estimated matrix $m_T \Sigma_T$ approaches the true matrix $A_0$ more closely as more questions are asked.

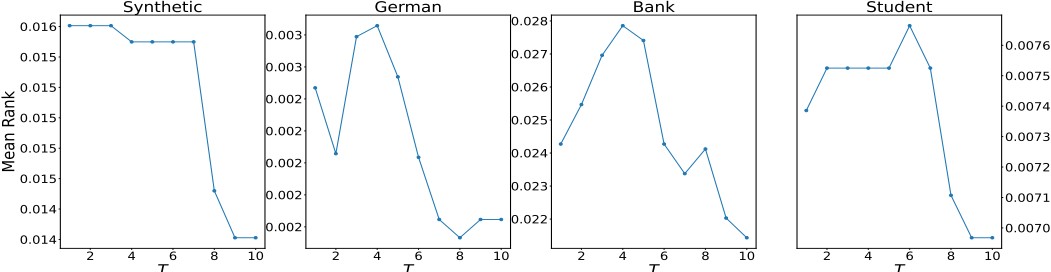

Figure 2: Effect of the number of questions $T$ to the average mean rank on all datasets.

## 7 CONCLUSION

In this paper, we propose a personalized algorithmic recourse framework based on Bayesian preference elicitation. Our approach employs mutual-information-maximization question-answering and efficient posterior updates for precise cost parameter estimation. We enhance recommendations with a graph-based sequential guidance method guided by the posterior distribution, achieving relevant and effective personalized recourse. Extensive evaluations of synthetic and real-world data validate the efficacy of our method in capturing individual preferences and delivering personalized solutions. One limitation is our assumption about a specific class of cost functions, which may not cover the diversity of individual characteristics. Future research should explore relaxing this assumption to accommodate a broader range of cost functions, considering factors like risk-seeking and risk-averse behaviors, to enhance the applicability of our framework.

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

The appendix is organized as follows.

- The discussion of listwise comparison is included in Section A.
- The proof details of Proposition 3.1 and Theorem 3.2 is given in Section B.
- The proof details of Proposition 4.3, Lemma 4.4, Lemma 4.5 and the details of algorithm are collected in Section C.
- Additional experiment results are included in Section D.

## A   EXPANDING FROM PAIRWISE TO LISTWISE COMPARISON

Consider the option of listwise comparison in addition to pairwise comparison. In listwise comparison, the objective is to identify a choice set comprising $r$ inputs that maximizes the mutual information between the response and the random matrix $\tilde{A}$. Consider a choice set $\mathbf{x}^i = (x_{i_1}, x_{i_2}, \ldots, x_{i_r})$ with $r$ elements. There are $\binom{N}{r}$ such possible choice sets. To facilitate the analysis, we introduce a permutation $\pi$, defined as $\pi : [r] \rightarrow [r]$, where $\pi(\mathbf{x}^j)_l$ denotes $x_{\pi(\mathbf{x}^j)_l}$.

One of the most widely used models for rank generation is the Plackett-Luce (PL) Model (Plackett, 1975; Luce, 2012). The PL model is defined as follows: Given a set of questions $\mathbf{x}^i = (x_{i_1}, x_{i_2}, \ldots, x_{i_r})$ and $A \in \mathbb{S}_+^d$, the probability of observing the response $\tilde{R}_i^\kappa$ as a certain ordered list $\pi(\mathbf{x}^i)$ is expressed as:

$$\text{Prob}\left(\tilde{R}_i^\kappa(A) = \pi(\mathbf{x}^i)\right) = \prod_{j=1}^r \frac{\exp(-\kappa c_A(\pi(\mathbf{x}^i)_j, x_0))}{\sum_{l=j}^r \exp(-\kappa c_A(\pi(\mathbf{x}^i)_l, x_0))}.$$

Notably, when $r = 2$, the PL model reduces to the BTL model as defined in equation (1).

Our objective is to identify a choice set $\mathbf{x}^i$ from $\mathcal{D}_1$ that maximizes the mutual information $MI(\tilde{A}, \tilde{R}_i^\kappa)$ between $\tilde{A}$ and $\tilde{R}_i$. Here,

$$MI(\tilde{A}, \tilde{R}_i^\kappa) = H(\tilde{R}_i^\kappa) - H(\tilde{R}_i^\kappa | \tilde{A}),$$

where $\Pi_r$ represents the set containing all permutations of $[r]$, with a cardinality of $r!$. $H(\tilde{R}_i^\kappa | \tilde{A})$ can be expressed as:

$$H(\tilde{R}_i^\kappa | \tilde{A}) = \int_{S \in \mathbb{S}_+^d} f_{\mathbb{P}_{t-1}}(S) H\left(\tilde{R}_i^\kappa | \tilde{A} = S\right) dS$$

$$= -\sum_{\pi \in \Pi_r} \int_{s \in \mathbb{S}_+^d} f_{\mathbb{P}_{t-1}}(S) \prod_{j=1}^r \frac{\exp(-\kappa c_S(\pi(\mathbf{x}^i)_j, x_0))}{\sum_{l=j}^r \exp(-\kappa c_S(\pi(\mathbf{x}^i)_l, x_0))} \log\left(\prod_{j=1}^r \frac{\exp(-\kappa c_S(\pi(\mathbf{x}^i)_j, x_0))}{\sum_{l=j}^r \exp(-\kappa c_S(\pi(\mathbf{x}^i)_l, x_0))}\right) dS.$$

Also, $H(\tilde{R}_i^\kappa)$ can be expressed as:

$$H(\tilde{R}_i) = -\sum_{\pi \in \Pi_r} \text{Prob}\left(\tilde{R}_i^\kappa(\tilde{A}) = \pi(\mathbf{x}^j)\right) \log\left(\text{Prob}\left(\tilde{R}_i^\kappa(\tilde{A}) = \pi(\mathbf{x}^j)\right)\right).$$

$\text{Prob}(\tilde{R}_i^\kappa(\tilde{A}) = \pi(\mathbf{x}^j))$ can be computed as:

$$\text{Prob}\left(\tilde{R}_i^\kappa(\tilde{A}) = \pi(\mathbf{x}^j)\right) = \int_{S \in \mathbb{S}_+^d} \text{Prob}\left(\tilde{R}_i^\kappa(\tilde{A}) = \pi(\mathbf{x}^j) | \tilde{A} = S\right) f_{\mathbb{P}_{t-1}}(S) dS$$

$$= \int_{S \in \mathbb{S}_+^d} \prod_{j=1}^r \frac{\exp(-\kappa c_S(\pi(\mathbf{x}^i)_j, x_0))}{\sum_{l=j}^r \exp(-\kappa c_S(\pi(\mathbf{x}^i)_l, x_0))} f_{\mathbb{P}_{t-1}}(S) dS.$$

To evaluate $MI(\tilde{A}, \tilde{R}_i^\kappa)$, we must compute $H(\tilde{R}_i^\kappa | \tilde{A})$ and $\text{Prob}(\tilde{R}_i^\kappa(\tilde{A}))$. Unfortunately, there are no straightforward analytical expressions for these two terms, necessitating the use of sampling methods for estimation (Gao et al., 2017; Mesner & Shalizi, 2020). Given that the complexity of computing $c_A(\cdot, \cdot)$ is $\mathcal{O}(d^2)$ and the complexity of estimating $H(\tilde{R}_i^\kappa | \tilde{A})$ and $\text{Prob}(\tilde{R}_i^\kappa(\tilde{A}))$ is

$\mathcal{O}(Md^2)$, where $M$ is the sample size, we further need to search for the choice set $\mathbf{x}^i$ with the highest mutual information. This search has a complexity of $\mathcal{O}(\binom{N}{r})$. Consequently, employing listwise comparison results in a substantial computational burden compared to pairwise comparison with an analytic expression for mutual information.

## B  PROOFS OF SECTION 3

### B.1  PROOF OF PROPOSITION 3.1

*Proof of Proposition 3.1.* The mutual information between $\tilde{A}$ and $\tilde{R}_{ij}^\kappa$ is given by

$$MI(\tilde{A}, \tilde{R}_{ij}^\kappa) = H(\tilde{R}_{ij}^\kappa) - H(\tilde{R}_{ij}^\kappa | \tilde{A}).$$

Using the definition of conditional entropy, $H(\tilde{R}_{ij}^\kappa | \tilde{A})$ can be calculated as follows:

$$
\begin{aligned}
H(\tilde{R}_{ij}^\kappa | \tilde{A}) &= \int_{S \in \mathbb{S}_+^d} f_{\mathbb{P}_{t-1}}(S) H(\tilde{R}_{ij}^\kappa | \tilde{A} = S) \mathrm{d}S \\
&= \int_{S \in \mathbb{S}_+^d} f_{\mathbb{P}_{t-1}}(S) h\left(\Phi(\kappa \Delta_{ij}(S))\right) \mathrm{d}S \\
&= \int_{S \in \mathbb{S}_+^d, \Delta_{ij}(S) \neq 0} f_{\mathbb{P}_{t-1}}(S) h\left(\Phi(\kappa \Delta_{ij}(S))\right) \mathrm{d}S + \int_{S \in \mathbb{S}_+^d, \Delta_{ij}(S) = 0} f_{\mathbb{P}_{t-1}}(S) h\left(\Phi(\kappa \Delta_{ij}(S))\right) \mathrm{d}S.
\end{aligned}
$$

Here, $h(z) = -z \log_2(z) - (1 - z) \log_2(1 - z)$. Since $h(\Phi(0)) = 1$, we have

$$\int_{S \in \mathbb{S}_+^d, \Delta_{ij}(S) = 0} f_{\mathbb{P}_{t-1}}(S) h\left(\Phi(\kappa \Delta_{ij}(S))\right) \mathrm{d}S = \int_{S \in \mathbb{S}_+^d, \Delta_{ij}(S) = 0} f_{\mathbb{P}_{t-1}}(S) \mathrm{d}S = \mathbb{P}_{t-1}(\Delta_{ij}(\tilde{A}) = 0).$$

Observe that $\Delta_{ij}(\tilde{A})$ is a continuous random variable defined on $\mathbb{R}$. Hence, $\mathbb{P}_{t-1}(\Delta_{ij}(\tilde{A}) = 0) = 0$ and

$$H(\tilde{R}_{ij}^\kappa | \tilde{A}) = \int_{S \in \mathbb{S}_+^d, \ \Delta_{ij}(S) \neq 0} f_{\mathbb{P}_{t-1}}(S) h\left(\Phi(\kappa \Delta_{ij}(S))\right) \mathrm{d}S.$$

It is evident that $0 < h(x) < 1$ for $0 < x < 1$, and as $x$ approaches 0 or 1, the limit of $h(x)$ is 0. Additionally, when $\Delta_{ij}(S) > 0$, the limit as $\kappa$ approaches infinity for $\Phi(\kappa \Delta_{ij}(S))$ is 1, while for $\Delta_{ij}(S) < 0$, the limit is 0. Therefore, when $\Delta_{ij}(S) \neq 0$,

$$\lim_{\kappa \to +\infty} h\left(\Phi(\kappa \Delta_{ij}(S))\right) = 0.$$

Also note that $|f_{\mathbb{P}_{t-1}}(S) h(\Phi(\kappa \Delta_{ij}(S))| \leq f_{\mathbb{P}_{t-1}}(S)$ and $\int_{S \in \mathbb{S}_+^d} f_{\mathbb{P}_{t-1}}(S) \mathrm{d}S = 1$. By Dominated convergence theorem, we can have

$$
\begin{aligned}
\lim_{\kappa \to +\infty} H(\tilde{R}_{ij}^\kappa | \tilde{A}) &= \lim_{\kappa \to \infty} \int_{S \in \mathbb{S}_+^d, \Delta_{ij}(S) \neq 0} f_{\mathbb{P}_{t-1}}(S) h\left(\Phi(\kappa \Delta_{ij}(S))\right) \mathrm{d}S \\
&= \int_{S \in \mathbb{S}_+, \Delta_{ij}(S) \neq 0} f_{\mathbb{P}_{t-1}}(S) \lim_{\kappa \to +\infty} h\left(\Phi(\kappa \Delta_{ij}(S))\right) \mathrm{d}S = 0.
\end{aligned}
$$

This can be explained by the fact that as $\kappa$ tends to infinity, $\tilde{R}_{ij}^\kappa$ becomes entirely determined by $\tilde{A}$, resulting in a conditional entropy of $\lim_{\kappa \to +\infty} H(\tilde{R}_{ij}^\kappa | \tilde{A}) = 0$.

As for $H(\tilde{R}_{ij}^\kappa)$, we have

$$H(\tilde{R}_{ij}^\kappa) = -\gamma_{ij}^\kappa \log_2(\gamma_{ij}^\kappa) - (1 - \gamma_{ij}^\kappa) \log_2(1 - \gamma_{ij}^\kappa).$$

The probability value $\gamma_{ij}^\kappa$ that $\tilde{R}_{ij}^\kappa$ admits the value of $+1$ is computed as

$$\gamma_{ij}^\kappa \triangleq \mathbb{P}(\tilde{R}_{ij}^\kappa(\tilde{A}) = +1)$$

$$= \int_{S \in \mathbb{S}_+^d} \mathbb{P}(\tilde{R}_{ij}^\kappa(\tilde{A}) = +1 | \tilde{A} = S) f_{\mathbb{P}_{t-1}}(S) \mathrm{d}S$$

$$= \int_{S \in \mathbb{S}_+^d} (1 - \Phi(\kappa \Delta_{ij}(S)) f_{\mathbb{P}_{t-1}}(S) \mathrm{d}S$$

$$= \int_{S \in \mathbb{S}_+^d, \Delta_{ij}(S) \neq 0} (1 - \Phi(\kappa \Delta_{ij}(S))) f_{\mathbb{P}_{t-1}}(S) \mathrm{d}S + \int_{S \in \mathbb{S}_+^d, \Delta_{ij}(S) = 0} (1 - \Phi(\kappa \Delta_{ij}(S))) f_{\mathbb{P}_{t-1}}(S) \mathrm{d}S.$$

Considering that $\int_{S \in \mathbb{S}_+^d, \Delta_{ij}(S)=0} (1 - \Phi(\kappa \Delta_{ij}(S)) f_{\mathbb{P}_{t-1}}(S) \mathrm{d}S = \frac{1}{2} \mathbb{P}_{t-1}(\Delta_{ij}(\tilde{A}) = 0)$ and based on Theorem B.2, we have $\mathbb{P}_{t-1}(\Delta_{ij}(\tilde{A}) = 0) = 0$. Therefore,

$$\gamma_{ij}^\kappa = \int_{S \in \mathbb{S}_+^d, \Delta_{ij}(S) \neq 0} (1 - \Phi(\kappa \Delta_{ij}(S))) f_{\mathbb{P}_{t-1}}(S) \mathrm{d}S.$$

For any $S \in \mathbb{S}_+^d$ such that $\Delta_{ij}(S) \neq 0$, we have

$$\lim_{\kappa \to +\infty} 1 - \Phi(\kappa \Delta_{ij}(S)) = \mathbb{I}_{\{\Delta_{ij}(S) \leq 0\}}(S).$$

Since $|(1 - \Phi(\kappa \Delta_{ij}(S)))f_{\mathbb{P}_{t-1}}(S)| \leq f_{\mathbb{P}_{t-1}}(S)$ and $\int_{S \in \mathbb{S}_+} f_{\mathbb{P}_{t-1}}(S) \mathrm{d}S = 1$, then by Dominated convergence theorem, we can see that

$$\gamma_{ij} \triangleq \lim_{\kappa \to \infty} \gamma_{ij}^\kappa$$

$$= \lim_{\kappa \to \infty, \Delta_{ij}(S) \neq 0} \int_{S \in \mathbb{S}_+^d} (1 - \Phi(\kappa \Delta_{ij}(S))) f_{\mathbb{P}_{t-1}}(S) \mathrm{d}S$$

$$= \int_{S \in \mathbb{S}_+^d, \Delta_{ij}(S) \neq 0} \lim_{\kappa \to \infty} (1 - \Phi(\kappa \Delta_{ij}(S))) f_{\mathbb{P}_{t-1}}(S) \mathrm{d}S$$

$$= \int_{S \in \mathbb{S}_+^d, \Delta_{ij}(S) \neq 0} \mathbb{I}_{\{\Delta_{ij}(S) \leq 0\}}(S) f_{\mathbb{P}_{t-1}}(S) \mathrm{d}S$$

$$= \mathbb{P}_{t-1}(\Delta_{ij}(\tilde{A}) \leq 0).$$

The last equality is because $\Delta_{ij}(\tilde{A})$ is a continuous variable. Thus, $\lim_{\kappa \to +\infty} H(\tilde{R}_{ij}^\kappa) = -\gamma_{ij} \log_2(\gamma_{ij}) - (1 - \gamma_{ij}) \log_2(1 - \gamma_{ij})$ and

$$\lim_{\kappa \to +\infty} MI(\tilde{R}_{ij}^\kappa, \tilde{A}) = \lim_{\kappa \to +\infty} H(\tilde{R}_{ij}^\kappa) - \lim_{\kappa \to +\infty} H(\tilde{R}_{ij}^\kappa | \tilde{A})$$

$$= -\gamma_{ij} \log_2(\gamma_{ij}) - (1 - \gamma_{ij}) \log_2(1 - \gamma_{ij}).$$

This finishes the proof. $\qquad\square$

## B.2 Proof of Theorem 3.2

Before proving Theorem 3.2, we present the following fact.

**Fact B.1** (Correlation). *Suppose that $A$ follows a Wishart distribution, $A \sim \mathcal{W}_d(m, \Sigma)$, for some parameters $(m, \Sigma)$ where $m \geq d$ is an integer. The followings hold:*

*(i) For any $u, v \in \mathbb{R}^d$, $\mathrm{Cov}(u^\top A u, v^\top A v) \geq 0$.*

*(ii) For any $u, v \in \mathbb{R}^d \backslash \{0\}$, let $\rho$ be the correlation coefficient between $u^\top A u$ and $v^\top A v$. Then $\rho = 1$ if and only if $u = kv$ for some $k \neq 0$.*

*Proof of Fact B.1.* We first prove part (i). Because $A$ follows a Wishart distribution with $m$ being an integer, we can decompose $A$ as a sum

$$A = \sum_{i=1}^m G_i G_i^\top,$$

where $G_1, \ldots, G_m$ are independent $d$-variate Gaussian random vectors with zero mean:

$$G_i = \left(g_i^1, g_i^2, \ldots, g_i^d\right) \sim \mathcal{N}_d(0, \Sigma).$$

Therefore, we find

$$
\begin{aligned}
\mathrm{Cov}\left(u^\top A u, v^\top A v\right) &= \sum_{i=1}^{m} \sum_{j=1}^{m} \mathrm{Cov}\left(u^\top G_i G_i^\top u, v^\top G_j G_j^\top v\right) \\
&= \sum_{i=1}^{m} \sum_{j=1}^{m} \mathrm{Cov}\left(\|G_i^\top u\|_2^2, \|G_j^\top v\|_2^2\right) \\
&= \sum_{i=1}^{m} \mathrm{Cov}\left(\|G_i^\top u\|_2^2, \|G_i^\top v\|_2^2\right).
\end{aligned}
$$

The last equality follows from that $G_i$ and $G_j$ are independent when $i \neq j$, thus $\|G_i^\top u\|_2^2$ and $\|G_i^\top v\|_2^2$ are independent and $\mathrm{Cov}(\|G_i^\top u\|_2^2, \|G_i^\top v\|_2^2) = 0$.

If $\mathrm{Cov}\left(\|G_i^\top u\|_2^2, \|G_i^\top v\|_2^2\right) \geq 0$, for $i = 1, 2, \ldots, n$, then $\mathrm{Cov}\left(u^\top A u, v^\top A v\right) \geq 0$. Since

$$
\begin{aligned}
\mathrm{Cov}\left(\|G_i^\top u\|_2^2, \|G_i^\top v\|_2^2\right) &= \mathrm{Cov}\left(\sum_{k=1}^{d}\left(u_k g_i^k\right)^2, \sum_{l=1}^{d}\left(v_l g_i^l\right)^2\right) \\
&= \sum_{k=1}^{d} \sum_{l=1}^{d} \mathrm{Cov}\left(\left(u_k g_i^k\right)^2, \left(v_l g_i^l\right)^2\right) \\
&= \sum_{k=1}^{d} \sum_{l=1}^{d} (u_k v_l)^2 \mathrm{Cov}\left(\left(g_i^k\right)^2, \left(g_i^l\right)^2\right).
\end{aligned}
$$

If $\mathrm{Cov}\left(\left(g_i^k\right)^2, \left(g_i^l\right)^2\right) \geq 0$, for $k, l \in [d]$, $\mathrm{Cov}\left(\|G_i^\top u\|_2^2, \|G_i^\top v\|_2^2\right) \geq 0$. If $k = l$, then

$$\mathrm{Cov}\left(\left(g_i^k\right)^2, \left(g_i^k\right)^2\right) = \mathrm{Var}\left(\left(g_i^k\right)^2\right) \geq 0.$$

Now we consider the case $k \neq l$. Without loss of generality, we assume that $k < l$. Since $G_i$ follows a $d$-variate normal distribution with zero mean, $\left(g_i^k, g_i^l\right)$ follows a bivariate normal distribution with zero mean:

$$\left(g_i^k, g_i^l\right) \sim \mathcal{N}_2\left(0, \begin{pmatrix} \Sigma_{kk} & \Sigma_{kl} \\ \Sigma_{lk} & \Sigma_{ll} \end{pmatrix}\right).$$

To simplify the proof, we temporarily drop the $i$ from $\left(g_i^k, g_i^l\right)$ and let $\sigma_1^2 = \Sigma_{kk}$, $\sigma_2^2 = \Sigma_{ll}$ and the correlation coefficient of $g_i^k$ and $g_i^l$ be $\rho$. Then, $\Sigma_{kl} = \Sigma_{lk} = \rho \sigma_1 \sigma_2$.

$$
\begin{aligned}
\mathrm{Cov}\left(\left(g^k\right)^2, \left(g^l\right)^2\right) &= \mathbb{E}\left[\left(g^k\right)^2 \left(g^l\right)^2\right] - \mathbb{E}\left[\left(g^k\right)^2\right] \mathbb{E}\left[\left(g^l\right)^2\right] \\
&= \mathbb{E}\left[\left(g^k\right)^2 \left(g^l\right)^2\right] - \mathrm{Var}\left[g^k\right] \mathrm{Var}\left[g^l\right] \\
&= \mathbb{E}\left[\left(g^k\right)^2 \left(g^l\right)^2\right] - \sigma_1^2 \sigma_2^2.
\end{aligned}
\tag{8}
$$

It remains to compute $\mathbb{E}\left[\left(g^k\right)^2 \left(g^l\right)^2\right]$. Based on the law of total expectation, we have

$$\mathbb{E}\left[\left(g^k\right)^2 \left(g^l\right)^2\right] = \mathbb{E}\left[\mathbb{E}\left[\left(g^k\right)^2 \left(g^l\right)^2 \Big| g^k\right]\right] = \mathbb{E}\left[\left(g^k\right)^2 \mathbb{E}\left[\left(g^l\right)^2 \Big| g^k\right]\right]. \tag{9}$$

Based on Ross (2010, Example 5c in Chapter 6), the distribution of $g^l$ conditional on $g^k = a$ follows a normal distribution

$$g^l | g^k = a \sim \mathcal{N}\left(\rho \frac{\sigma_2}{\sigma_1} a, (1 - \rho^2)\sigma_2^2\right).$$

Thus, we find

$$\mathbb{E}\left[\left(g^l\right)^2\Big|g^k = a\right] = \mathrm{Var}\left[g^l|g^k = a\right] + \mathbb{E}\left[g^l|g^k = a\right]^2 = (1-\rho^2)\sigma_2^2 + \left(\rho\frac{\sigma_2}{\sigma_1}a\right)^2.$$

Then we have

$$\mathbb{E}\left[\left(g^l\right)^2\Big|g^k\right] = (1-\rho^2)\sigma_2^2 + \left(\rho\frac{\sigma_2}{\sigma_1}g^k\right)^2. \tag{10}$$

Plugging (10) into (9), we can have

$$\begin{aligned}
\mathbb{E}\left[\left(g^k\right)^2\left(g^l\right)^2\right] &= \mathbb{E}\left[\left((1-\rho^2)\sigma_2^2 + \left(\rho\frac{\sigma_2}{\sigma_1}g^k\right)^2\right)\left(g^k\right)^2\right] \\
&= (1-\rho^2)\sigma_2^2\mathbb{E}\left[\left(g^k\right)^2\right] + \left(\rho\frac{\sigma_2}{\sigma_1}\right)^2\mathbb{E}\left[\left(g^k\right)^4\right] \\
&= (1-\rho^2)\sigma_2^2\sigma_1^2 + \left(\rho\frac{\sigma_2}{\sigma_1}\right)^2\left(3\sigma_1^4\right) \\
&= \sigma_1^2\sigma_2^2 + 2\rho^2\sigma_1^2\sigma_2^2.
\end{aligned} \tag{11}$$

Substituting (11) into (8), we obtain

$$\mathrm{Cov}\left(\left(g^k\right)^2, \left(g^l\right)^2\right) = \mathbb{E}\left[\left(g^k\right)^2\left(g^l\right)^2\right] - \sigma_1^2\sigma_2^2 = 2\rho^2\sigma_1^2\sigma_2^2 \geq 0.$$

This completes the proof for part (i). To prove part (ii), note that the Kronecker form of $\mathrm{Cov}(u^\top A u, v^\top A v)$ is

$$\mathrm{Cov}(u^\top A u, v^\top A v) = (u \otimes u)^\top \mathrm{Cov}(\mathrm{vec}(A))(v \otimes v).$$

Here, $\mathrm{Cov}(\mathrm{vec}(A)) = m[\Sigma \otimes \Sigma][I_{d^2} + C]$ (Christensen, 2015) and $C$ is the commutation matrix, defined as

$$C = \sum_{k=1}^{d}\sum_{l=1}^{d}(e_l \otimes e_k)(e_k \otimes e_l)^\top.$$

According to Christensen (2015), the matrix $[\Sigma \otimes \Sigma][I_{d^2} + C]$ is positive definite if $\Sigma$ is positive definite. Since we have $\Sigma$ is positive definite, then $\mathrm{Cov}(\mathrm{vec}(A)) = m[\Sigma \otimes \Sigma][I_{d^2} + C]$ is also positive definite. Then there exist $B \in \mathbb{S}_{++}^{d^2}$ such that $\mathrm{Cov}(\mathrm{vec}(A)) = BB$. The correlation coefficient $\rho$ is computed as follows

$$\begin{aligned}
\rho &= \frac{\mathrm{Cov}(u^\top A u, v^\top A v)}{\sqrt{\mathrm{Var}(u^\top A u)\mathrm{Var}(u^\top A u)}} \\
&= \frac{(u \otimes u)^\top \mathrm{Cov}(\mathrm{vec}(A))(v \otimes v)}{\sqrt{(u \otimes u)^\top \mathrm{Cov}(\mathrm{vec}(A))(u \otimes u)(v \otimes v)^\top \mathrm{Cov}(\mathrm{vec}(A))(v \otimes v)}} \\
&= \frac{(B(u \otimes u))^\top (B(v \otimes v))}{\|B(u \otimes u)\|_2\|B(v \otimes v)\|_2}.
\end{aligned}$$

By Cauchy-Schwarz inequality, $\rho = 1$ if and only if $B(u \otimes u) = \lambda B(v \otimes v)$, $\lambda > 0$. Since $B \in \mathbb{S}_{++}^{d^2}$, we have

$$B(u \otimes u) = \lambda B(v \otimes v) \Leftrightarrow u \otimes u = \lambda(v \otimes v).$$

Note that $u \otimes u = \mathrm{vec}(uu^\top)$. Thus, $uu^\top = \lambda vv^\top$ and $u^\top u = \lambda v^\top v$. Consider that

$$\lambda\|v\|_2^2\|u\|_2^2 = \|u\|_2^4 = u^\top uu^\top u = u^\top(\lambda vv^\top)u = \lambda(v^\top u)^2.$$

Therefore, we have $(v^\top u)^2 = \|v\|_2^2\|u\|_2^2$. By Cauchy-Schwarz inequality, we have $u = kv$, for $k \in \mathbb{R}$. Since $u, v \in \mathbb{R}^d\backslash\{0\}$, we have $k \neq 0$. $\square$

The proof of Theorem 3.2 relies on the following auxiliary result which asserts the distributional form of the difference of two Gamma random variables.

**Theorem B.2** (Difference of correlated Gamma distributions (Holm & Alouini, 2004, Theorem 6))**.** *Let $Z_1 \sim \Gamma(\alpha, \beta_1)$ and $Z_2 \sim \Gamma(\alpha, \beta_2)$ be two correlated Gamma random variables, where $(\alpha, \beta)$ is the shape-scale parameterization. Suppose that the correlation coefficient $\rho = \mathrm{Cov}(Z_1, Z_2)/\sqrt{\mathrm{Var}(Z_1)\mathrm{Var}(Z_2)} \in [0, 1)$. The difference $\Delta = Z_1 - Z_2$ has the probability density function*

$$\forall \delta \in \mathbb{R} \backslash \{0\}: \quad f_\Delta(\delta) = \frac{|\delta|^{\alpha+\frac{1}{2}}}{\Gamma(\alpha)\sqrt{\pi}\sqrt{\beta_1\beta_2}(1-\rho)} \left( \frac{1}{(\beta_1+\beta_2)^2 + -4\beta_1\beta_2\rho} \right)^{\frac{2\alpha-1}{4}}$$
$$\exp\left( \frac{\frac{\delta}{2}}{1-\rho} \left( \frac{1}{\beta_2} - \frac{1}{\beta_1} \right) \right) K_{\alpha-\frac{1}{2}} \left( |\delta| \frac{\sqrt{(\beta_1+\beta_2)^2 - 4\beta_1\beta_2\rho}}{2\beta_1\beta_2(1-\rho)} \right),$$

*where $K_{\alpha-\frac{1}{2}}$ denotes the modified Bessel function of the second kind and order $\alpha - \frac{1}{2}$.*

We are now ready to prove Theorem 3.2.

*Proof of Theorem 3.2.* Because $\mathbb{P} \sim \mathcal{W}_d(m, \Sigma)$, let $\sigma_i^2 = (x_i - x_0)^\top \Sigma(x_i - x_0)$, we have $(x_i - x_0)^\top A(x_i - x_0) \sim \sigma_i^2 \chi_m^2$ or equivalently $(x_i - x_0)^\top A(x_i - x_0) \sim \Gamma(k = \frac{m}{2}, \theta = 2\sigma_i^2)$, where $(k, \theta)$ is the shape-scale parameter of the Gamma distribution. Similarly, we have $(x_j - x_0)^\top A(x_j - x_0) \sim \Gamma(k = \frac{m}{2}, \theta = 2\sigma_j^2)$. The correlation of $(x_i - x_0)^\top A(x_i - x_0)$ and $(x_j - x_0)^\top A(x_j - x_0)$ can be computed as

$$\rho = \frac{\mathrm{Cov}((x_i - x_0)^\top A(x_i - x_0), (x_j - x_0)^\top A(x_j - x_0))}{\sqrt{\mathrm{Var}((x_i - x_0)^\top A(x_i - x_0))\mathrm{Var}((x_j - x_0)^\top A(x_j - x_0))}}.$$

By the property of Chi-square distribution, we have $\mathrm{Var}((x_i - x_0)^\top A(x_i - x_0)) = 2m\sigma_i^4$ and $\mathrm{Var}((x_j - x_0)^\top A(x_j - x_0)) = 2m\sigma_j^4$. Let $z_i = x_i - x_0$ and $z_j = x_j - x_0$. The covariance is calculated as follows:

$$\mathrm{Cov}(z_i^\top A z_i, z_j^\top A z_j) = \mathrm{Cov}\Big( \sum_{k_1,k_2=1}^{d} A_{k_1,k_2} z_{i,k_1} z_{i,k_2}, \sum_{l_1,l_2=1}^{d} A_{l_1,l_2} z_{j,l_1} z_{j,l_2} \Big)$$
$$= \sum_{k_1,k_2=1}^{d} \sum_{l_1,l_2=1}^{d} z_{i,k_1} z_{i,k_2} z_{j,l_1} z_{j,l_2} \mathrm{Cov}(A_{k_1,k_2}, A_{l_1,l_2}).$$

According to Christensen (2015), the covariance matrix of $\mathbb{P} \sim \mathcal{W}_d(\Sigma, m)$ has a Kronecker form, i.e. $\mathrm{Cov}(\mathrm{Vec}(A)) = m[\Sigma \otimes \Sigma][I_{d^2} + C]$. Using this result, we can write $\mathrm{Cov}(z_i^\top A z_i, z_j^\top A z_j) = m[z_i \otimes z_i]^\top[\Sigma \otimes \Sigma][I_{d^2} + C][z_j \otimes z_j]$. Thus, we have

$$\rho = \frac{[z_i \otimes z_i]^\top[\Sigma \otimes \Sigma][I_{d^2} + C][z_j \otimes z_j]}{2\sigma_i^2\sigma_j^2}.$$

Let $\Delta = z_i^\top A z_i - z_j^\top A z_j$. Considering Fact B.1, where $z_i$ and $z_j$ are non-parallel vectors, we can establish that $\rho \in [0, 1)$, allowing us to apply Theorem B.2. This theorem enables us to determine that the distribution of $\Delta$ adheres to the type II McKay distribution (Holm & Alouini, 2004), characterized by the following parameters:

$$a = \frac{m-1}{2}, \ b = \frac{8\sigma_i^2\sigma_j^2(1-\rho)}{\sqrt{4(\sigma_i^2-\sigma_j^2)^2 + 16\sigma_i^2\sigma_j^2(1-\rho)}}, \ c = -\frac{2(\sigma_i^2-\sigma_j^2)}{\sqrt{4(\sigma_i^2-\sigma_j^2)^2 + 16\sigma_i^2\sigma_j^2(1-\rho)}}.$$

Let $E_0 = \frac{(1-c^2)^{a+\frac{1}{2}}}{\sqrt{\pi}2^a b^{a+1}\Gamma(a+\frac{1}{2})}$. The probability $\gamma_{ij} = \mathbb{P}(\langle A, M_{ij} \rangle \leq 0)$ can be expressed as follows:

$$\gamma_{ij} = \mathbb{P}(\langle A, M_{ij} \rangle \leq 0) = \int_{-\infty}^{0} f_\Delta(\delta) \mathrm{d}\delta$$
$$= E_0 \int_{-\infty}^{0} |\delta|^a \exp\left( -\delta\frac{c}{b} \right) K_a\left( \frac{|\delta|}{b} \right) \mathrm{d}\delta$$
$$= E_0 \int_{0}^{\infty} \delta^a \exp\left( \delta\frac{c}{b} \right) K_a\left( \frac{\delta}{b} \right) \mathrm{d}\delta.$$

The solution of $\int_0^\infty \delta^a \exp(\delta \frac{c}{b}) K_a(\frac{\delta}{b}) \mathrm{d}\delta$ can be found in Holm & Alouini (2004, Appendix 2):

$$\int_0^\infty x^{\mu-1} \exp(-\eta x) K_a(\beta x) \mathrm{d}x = \frac{\sqrt{\pi}(2\beta)^a}{(\eta+\beta)^{\mu+a}} \frac{\Gamma(\mu+a)\Gamma(\mu-a)}{\Gamma(\mu+\frac{1}{2})}$$
$$_2F_1\left(\mu+a, a+\frac{1}{2}; \mu+\frac{1}{2}; \frac{\eta-\beta}{\eta+\beta}\right),$$

where $\mu > |a|$ and $\eta + \beta > 0$. To compute the integral, we can assign the values $\mu = a + 1$, $\eta = -\frac{c}{b}$, and $\beta = \frac{1}{b}$. Then we verify if they satisfy the required conditions. Since $m \geq 2$, $a$ is always positive, which means $\mu = a + 1 > |a|$. The condition $0 \leq \rho < 1$ implies that $b > 0$. Additionally, we have $0 < |c| < 1$. Therefore, we can conclude that $\eta + \beta = -\frac{c}{b} + \frac{1}{b} = \frac{1-c}{b} > 0$. Hence, the conditions for $\mu$, $\eta$, and $\beta$ are satisfied.

Solving the integral, we obtain:

$$\int_0^\infty \delta^{(a+1)-1} \exp\left(-\left(-\frac{c}{b}\right)\delta\right) K_a\left(\frac{1}{b}\delta\right) \mathrm{d}\delta = \frac{\sqrt{\pi}(2\frac{1}{b})^a}{(\frac{-c+1}{b})^{2a+1}} \frac{\Gamma(2a+1)\Gamma(1)}{\Gamma(a+\frac{3}{2})}$$
$$_2F_1\left(2a+1, a+\frac{1}{2}; a+\frac{3}{2}; \frac{-c-1}{-c+1}\right).$$

Plugging the result back, we obtain:

$$\gamma_{ij} = \left(\frac{1+c}{1-c}\right)^{a+\frac{1}{2}} \frac{\Gamma(2a+1)\Gamma(1)}{\Gamma(a+\frac{3}{2})\Gamma(a+\frac{1}{2})} \, _2F_1\left(2a+1, a+\frac{1}{2}; a+\frac{3}{2}; -\frac{1+c}{1-c}\right).$$

This completes the proof. $\qquad\square$

## C  DETAILS OF ALGORITHM AND PROOFS OF SECTION 4

### C.1  PROOF OF PROPOSITION 4.3

*Proof of Proposition 4.3.* Observe that the identity matrix $I$ is a feasible solution to problem (4). The minimal solution of problem (4) is determined only by the feasible decisions $\Sigma$ with an objective value of at most $\ell(I)$. All such solutions $\Sigma$ should satisfy

$$-\frac{m_{t-1}}{m} \log\det(\Sigma) + \mathrm{Tr}[(\Sigma_{t-1}^{-1} + \tau\kappa R_{ij}M_{ij})\Sigma] = \ell(\Sigma) \leq \ell(I) = \mathrm{Tr}[(\Sigma_{t-1}^{-1} + \tau\kappa R_{ij}M_{ij})],$$

where the first equality follows from the definition of $\ell$, while the second equality follows because $\log\det(I) = 0$. Rearranging the above inequality, we have

$$-\frac{m_{t-1}}{m} \log\det(\Sigma) \leq \mathrm{Tr}[(\Sigma_{t-1}^{-1} + \tau\kappa R_{ij}M_{ij})(I-\Sigma)]$$
$$\leq \|\Sigma_{t-1}^{-1} + \tau\kappa R_{ij}M_{ij}\|_F \|I-\Sigma\|_F$$
$$\leq \|\Sigma_{t-1}^{-1} + \tau\kappa R_{ij}M_{ij}\|_F \left(\|I\|_F + \|\Sigma\|_F\right)$$
$$\leq \|\Sigma_{t-1}^{-1} + \tau\kappa R_{ij}M_{ij}\|_F (\sqrt{d}+d),$$

where the second inequality follows from Cauchy-Schwarz, and the last inequality follows from that denoting by $\{\lambda_i\}_{i\leq d}$ the eigenvalues of the matrix $\Sigma$, we have

$$\|\Sigma\|_F = \sqrt{\mathrm{Tr}[\Sigma\Sigma]} = \sqrt{\sum_{i=1}^d \lambda_i^2} \leq \sqrt{\left(\sum_{i=1}^d \lambda_i\right)^2} = d.$$

Since $\sum_{i=1}^d \lambda_i = d$ and $\lambda_i > 0$, therefore $\lambda_i \leq d$, $i = 1, 2, \ldots d$. Setting $\lambda_{\min} = \min_{1\leq i\leq d} \lambda_i$, we have

$$-\frac{m_{t-1}}{m} \log(d^{d-1}\lambda_{\min}) \leq -\frac{m_{t-1}}{m} \log(\Pi_{i=1}^d \lambda_i) = -\frac{m_{t-1}}{m} \log\det(\Sigma) \leq \|\Sigma_{t-1}^{-1} + \tau\kappa R_{ij}M_{ij}\|_F (\sqrt{d}+d).$$

By rearranging the inequality above, we have $\lambda_{\min} \geq \varepsilon$, which implies $\Sigma \succeq \varepsilon I$. The statement above shows that adding the extra constraint $\Sigma \succeq \varepsilon I$ has no impact on the optimal value and the optimal solution of problem (4) and we can transform problem (4) into the equivalent form (5). $\qquad\square$

## C.2   PROOF OF LEMMA 4.4

*Proof of Lemma 4.4.* We have

$$\|S - \Sigma\|_F^2 = \|V\operatorname{diag}(s)V^\top - \Sigma\|_F^2 = \|\operatorname{diag}(s) - V^\top \Sigma V\|_F^2,$$

thus, we can rewrite the projection operator as

$$\operatorname{Proj}(S) = \arg\min\{\|\operatorname{diag}(s) - V^\top \Sigma V\|_F^2 : \Sigma \in \mathbb{S}_+^d, \ \Sigma \succeq \varepsilon I, \ \operatorname{Tr}[\Sigma] = d\}.$$

Let $\Sigma^\star$ be the minimizer of the above optimization problem. We can observe that the minimum occurs when $(V^\top \Sigma^\star V)_{ij} = 0$, $i \neq j$, therefore $V^\top \Sigma^\star V$ is diagonal. In other words, $\Sigma^\star$ shares the same eigenbasis with $S$ and we can parametrize $\Sigma^\star = V\operatorname{diag}(\sigma^\star)V^\top$, where $\sigma^\star \in \mathbb{R}^d$ is a vector containing the eigenvalues of $\Sigma^\star$. Moreover, $\sigma^\star$ solves

$$\begin{aligned} \min \quad & \|\sigma - s\|_2^2 \\ \text{s.t.} \quad & \sigma \in \mathbb{R}_+^d, \ \sigma \geq \varepsilon, \ \sigma^\top \mathbf{1} = d, \end{aligned} \tag{12}$$

in which the constraints of the above vector-optimization problem are obtained by rewriting the corresponding semidefinite constraints $\Sigma \succeq \varepsilon I$ and $\operatorname{Tr}[\Sigma] = d$ using the eigenvalues.

Problem (12) is similar to the projection onto the simplex problem (6), which is recited here for convenience:

$$\begin{aligned} \min \quad & \|\lambda - (s - \varepsilon\mathbf{1})\|_2^2 \\ \text{s.t.} \quad & \lambda \geq 0, \ \lambda^\top \mathbf{1} = (1 - \varepsilon)d. \end{aligned}$$

Let $\lambda^\star$ be the optimal solution to (6), then the following relationship holds: $\sigma^\star = \lambda^\star + \varepsilon\mathbf{1}$. To see this, note that $\lambda^\star + \varepsilon\mathbf{1}$ is a feasible solution to (12) and $\sigma^\star - \varepsilon\mathbf{1}$ is a feasible solution to (6). We have

$$\|\sigma^\star - s\|_2^2 \leq \|\lambda^\star + \varepsilon\mathbf{1} - s\|_2^2 \leq \|\sigma^\star - \varepsilon\mathbf{1} - (s - \varepsilon\mathbf{1})\|_2^2 = \|\sigma^\star - s\|_2^2.$$

Thus, $\lambda^\star + \varepsilon\mathbf{1}$ is also an optimal solution to (12). Since the set $\{\lambda : \lambda \geq \varepsilon, \lambda^\top \mathbf{1} = d\}$ is convex and closed, the projection onto this set is unique. Therefore, $\sigma^\star = \lambda^\star + \varepsilon\mathbf{1}$. This completes the proof. □

## C.3   PROOF OF LEMMA 4.5

*Proof of Lemma 4.5.* The Hessian of $\ell$ is

$$\nabla^2 \ell(\Sigma) = \frac{m_{t-1}}{m} \Sigma^{-1} \otimes \Sigma^{-1}.$$

For any $\Sigma \in D$, its eigenvalues lie in the interval $[\varepsilon, d]$. Consider the eigendecomposition of $\Sigma = U\Lambda U^\top$, where $\Lambda$ is a diagonal matrix containing the eigenvalues of $\Sigma$. We have

$$\begin{aligned} \nabla^2 \ell(\Sigma) &= \frac{m_{t-1}}{m} (U\Lambda^{-1}U^\top) \otimes (U\Lambda^{-1}U^\top) \\ &= \frac{m_{t-1}}{m} (U \otimes U)(\Lambda^{-1} \otimes \Lambda^{-1})(U^\top \otimes U^\top). \end{aligned}$$

Observe that

$$\frac{1}{d^2} I \preceq \Lambda^{-1} \otimes \Lambda^{-1} \preceq \frac{1}{\varepsilon^2} I.$$

Thus,

$$\frac{m_{t-1}}{md^2} I \preceq \nabla^2 \ell(\Sigma) \preceq \frac{m_{t-1}}{m\varepsilon^2} I.$$

Therefore, $\ell$ is strongly convex and has Lipschitz gradient on $D$. □

## C.4   DETAILS OF ALGORITHM

Consider the simplex $\{\lambda \in \mathbb{R}^d : \lambda \geq 0, \lambda^\top \mathbf{1} = (1 - \varepsilon)d\}$, where $\varepsilon$ is a small positive value. Building on previous work by Held et al. (1974), we present Algorithm 3 for efficiently computing the projection onto the simplex. The algorithm has a complexity of $\mathcal{O}(d \log d)$.

---

**Algorithm 2** Projection onto $D$

---

**Input:** Input $S \in \mathbb{S}^d$, $\varepsilon$
    Conduct eigendecomposition of $S = V\Lambda V^\top$.
    $\lambda \leftarrow \text{diagonal}(\Lambda)$, $y \leftarrow \lambda - \varepsilon\mathbf{1}$.
    Obtain the projection $y^\star$ of $y$ onto simplex using Algorithm 3.
    $\lambda^\star \leftarrow y^\star + \varepsilon\mathbf{1}$, $\Lambda^\star \leftarrow \text{diag}(\lambda^\star)$, $\Sigma \leftarrow V\Lambda^\star V^\top$.
**Output:** $\Sigma$

---

**Algorithm 3** Projection onto simplex $\{\lambda \in \mathbb{R}^d : \lambda \geq 0, \lambda^\top \mathbf{1} = (1-\varepsilon)d\}$

---

**Input:** $x \in \mathbb{R}^d$
    Sort $x$ into $u$: $u_1 \geq u_2 \geq \cdots \geq u_d$.
    Set $M = \max_{1 \leq k \leq d} \left\{ k : \left( \sum_{i=1}^k u_i - (1-\varepsilon)d \right)/k < u_k \right\}$.
    Set $\eta = \left( \sum_{i=1}^M u_i - (1-\varepsilon)d \right)/M$.
    For $i = 1, 2, \ldots, d$, set $z_i = \max\{x_i - \eta, 0\}$.
**Output:** $z$

---

# D  ADDITIONAL EXPERIMENTS

## D.1  DATASET

We conduct our experiments on three real-world datasets: German (Dua & Graff, 2017), Bank (Dua & Graff, 2017), and Student (Cortez & Silva, 2008) dataset. Table 3 reports the features of each dataset. For synthetic dataset, we sample a tuple $(x_1, x_2)$ from a uniform distributions $\mathcal{U}(-2, 4)$ and $\mathcal{U}(-2, 7)$, respectively. Subsequently, we compute $f(x_1) = 1 + x_1 + 2x_1^2 + x_1^3 - x_1^4$. We set the label $y = 1$ if $x_2 \geq f(x_1)$, otherwise we set $y = 0$.

Table 3: The features of real-world datasets.

| Dataset | Features |
|---------|----------|
| German | status, duration, credit amount, personal status, age |
| Bank | age, education, balance, housing, loan, campaign, previous, outcome |
| Student | age, study time, famsup, higher, internet, health, absence, G1, G2 |

## D.2  CLASSIFIER

In this section, we report the accuracy and AUC of the MLP classifiers across the four datasets in Table 4.

Table 4: Accuracy and AUC of the MLP classifier.

| Dataset | Synthetic | German | Bank | Student |
|---------|-----------|--------|------|---------|
| Accuracy | 0.97 | 0.72 | 0.88 | 0.93 |
| AUC | 0.99 | 0.62 | 0.66 | 0.97 |

## D.3  COMPARISON WITH GRADIENT-BASED METHOD

We have conducted a comparative analysis between our proposed Bayesian PR method and the non-graph-based methods, Wachter and DiCE. The results, as presented in Table 5 and Table 6, reveal that our proposed approach demonstrates comparable performance to the gradient-based methods while achieving superior results on real-world datasets such as *German*, *Bank*, and *Student*.

Table 5: Comparison of cost and validity between Bayesian PR and non-graph-based method DiCE and Wachter using Mahalanobis distance as the true cost.

| Method | Synthetic | | German | | Bank | | Student | |
|---|---|---|---|---|---|---|---|---|
| | Cost | Validity | Cost | Validity | Cost | Validity | Cost | Validity |
| DiCE | $0.01 \pm 0.01$ | $1.00 \pm 0.00$ | $0.17 \pm 0.19$ | $1.00 \pm 0.00$ | $2.35 \pm 0.45$ | $1.00 \pm 0.00$ | $0.44 \pm 0.57$ | $1.00 \pm 0.00$ |
| Wachter | $\mathbf{0.002} \pm 0.004$ | $1.00 \pm 0.00$ | $0.11 \pm 0.08$ | $1.00 \pm 0.00$ | $0.55 \pm 0.28$ | $1.00 \pm 0.00$ | $0.41 \pm 0.28$ | $1.00 \pm 0.00$ |
| Bayesian PR | $0.09 \pm 0.06$ | $1.00 \pm 0.00$ | $\mathbf{0.09} \pm 0.78$ | $1.00 \pm 0.00$ | $\mathbf{0.54} \pm 0.31$ | $1.00 \pm 0.00$ | $\mathbf{0.39} \pm 0.23$ | $1.00 \pm 0.00$ |

Table 6: Comparison of cost and validity between Bayesian PR and non-graph-based method DiCE and Wachter using $\ell_1$ norm as the true cost.

| Method | Synthetic | | German | | Bank | | Student | |
|---|---|---|---|---|---|---|---|---|
| | Cost | Validity | Cost | Validity | Cost | Validity | Cost | Validity |
| DiCE | $0.10 \pm 0.04$ | $1.00 \pm 0.00$ | $0.21 \pm 0.22$ | $1.00 \pm 0.00$ | $3.23 \pm 1.15$ | $1.00 \pm 0.00$ | $0.53 \pm 0.95$ | $1.00 \pm 0.00$ |
| Wachter | $\mathbf{0.04} \pm 0.03$ | $1.00 \pm 0.00$ | $0.15 \pm 0.11$ | $1.00 \pm 0.00$ | $0.85 \pm 0.44$ | $1.00 \pm 0.00$ | $1.07 \pm 0.75$ | $1.00 \pm 0.00$ |
| Bayesian PR | $0.10 \pm 0.67$ | $1.00 \pm 0.00$ | $\mathbf{0.12} \pm 0.09$ | $1.00 \pm 0.00$ | $\mathbf{0.81} \pm 0.43$ | $1.00 \pm 0.00$ | $\mathbf{0.96} \pm 0.57$ | $1.00 \pm 0.00$ |

### D.4 Posterior Loss

Figure 3 shows a sample of loss $\ell(\Sigma)$ for each dataset. We perform the question selection for $T = 5$ sessions. Given the optimum degree of freedom $m$, we perform the posterior update for 1000 iterations and plot the results. The figure shows that the losses converge after 1000 iterations.

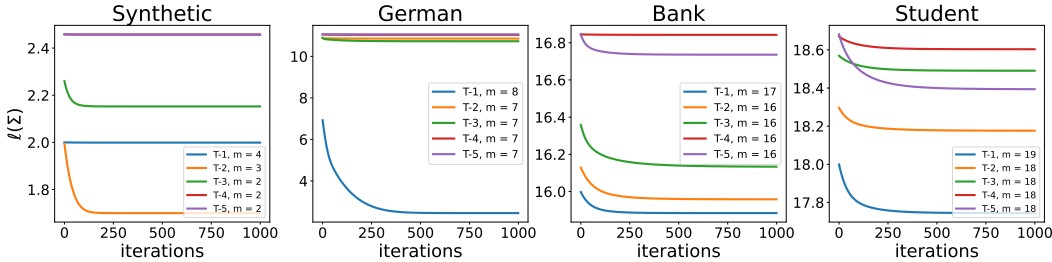

Figure 3: The posterior-update loss $\ell(\Sigma)$ given the optimum degree of freedom $m$ for each dataset. We plot the loss for different numbers of questions $T$.

### D.5 Risk Measurement

In Figures 4 to 7, we present histograms illustrating the Mahalanobis distances for each dataset based on the posterior distribution $\mathbb{P}_T$ when $T = 10$, with varying values of $\kappa$ ($+\infty$, 1, 5, and 10). For each pair of $x_0$ and $x_r$, we generate 1000 samples of $A$ from $\mathbb{P}_T$ and calculate the Mahalanobis distance between $x_0$ and $x_r$ for each sample. These histograms offer a visual representation of the distribution of these sampled distances. Additionally, we include statistics such as the mean, standard deviation, and the true Mahalanobis distance computed using $A_0$. Notably, as $T$ increases, the means tend to converge closer to the true cost values for all datasets and across all values of $\kappa$, accompanied by narrower standard deviations, demonstrating the effectiveness of our framework even with finite $\kappa$.

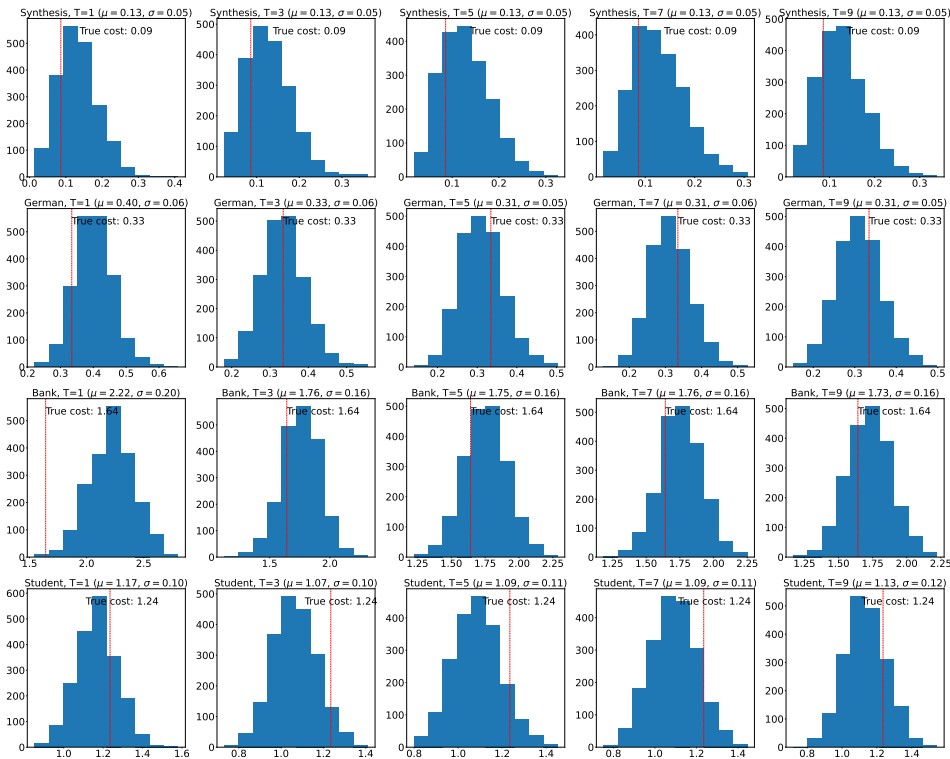

Figure 4: The histogram of Mahalanobis distances based on the posterior distribution for all datasets when $\kappa = +\infty$.

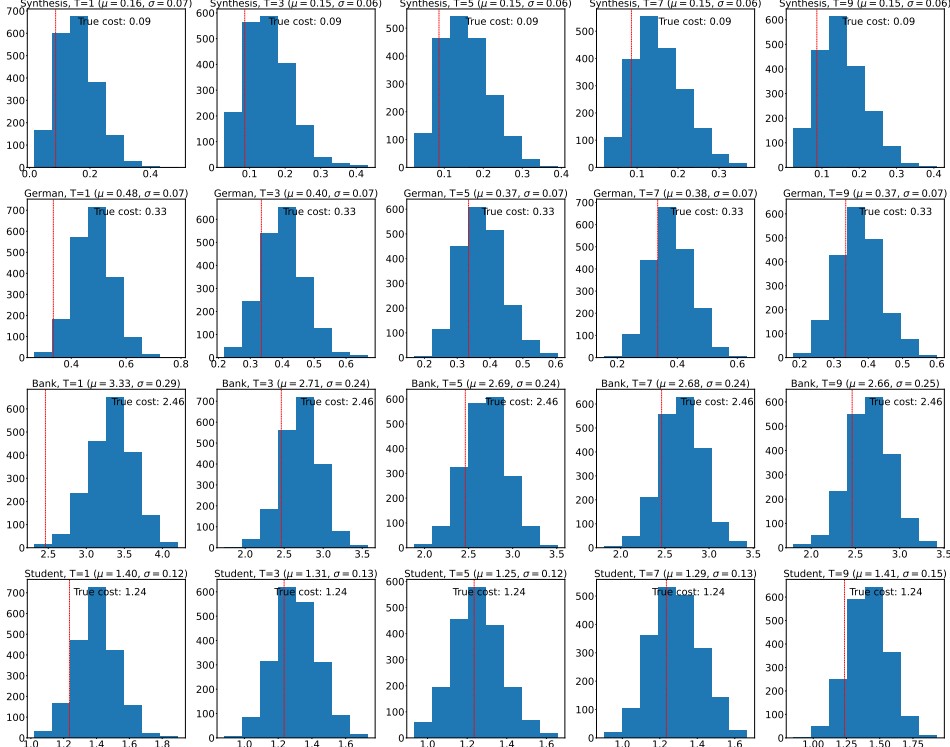

Figure 5: The histogram of Mahalanobis distances based on the posterior distribution for all datasets when $\kappa = 1$.

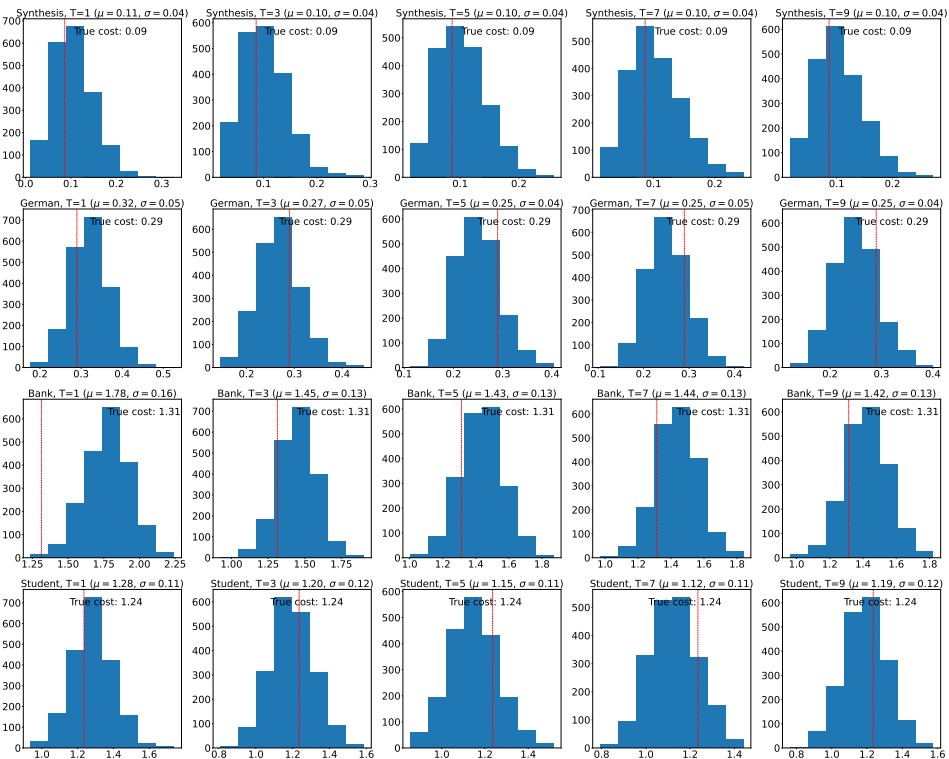

Figure 6: The histogram of Mahalanobis distances based on the posterior distribution for all datasets when $\kappa = 5$.

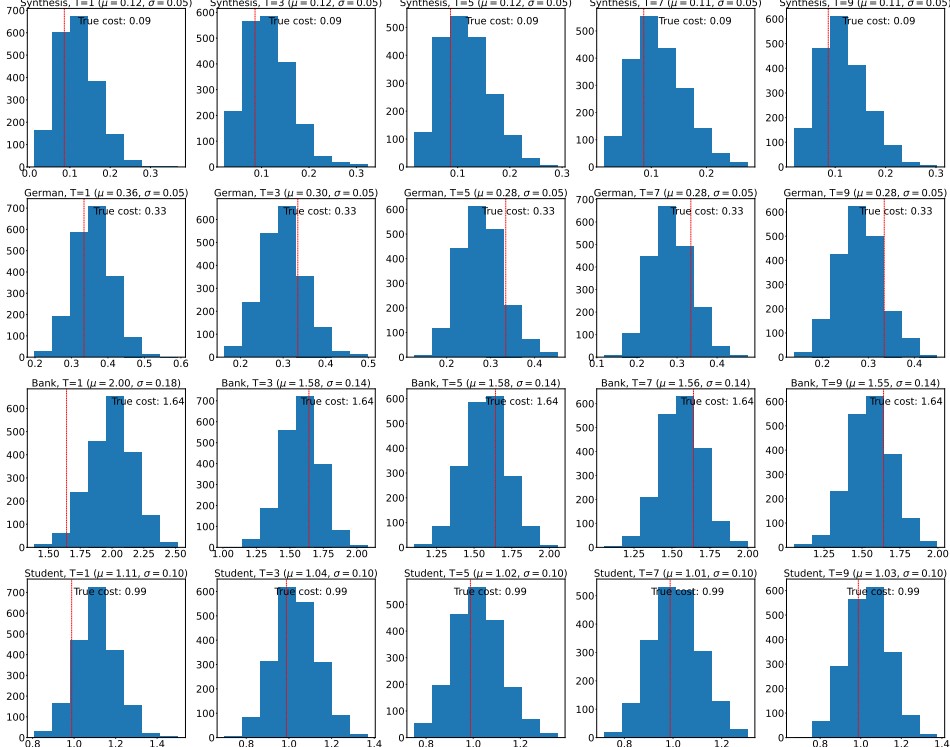

Figure 7: The histogram of Mahalanobis distances based on the posterior distribution for all datasets when $\kappa = 10$.

