# OpenReview forum: "Bayesian Preference Elicitation for Personalized Prefactual Recommendation"
_ICLR.cc/2024/Conference — Submitted to ICLR 2024_

### Official Review · Reviewer_yvPj · 2023-10-30

**Soundness:** 3 good
**Presentation:** 3 good
**Contribution:** 1 poor
**Rating:** 3
**Confidence:** 5

**Summary:**

The paper presents Bayesian PR, a method to generate prefactual recommendations by learning a personalized cost function by collecting pairwise feedback from a user. These recommendations lie in the algorithmic recourse field since they give actionable feedback to a user to overturn the prediction of a machine learning model. The authors provide a Bayesian framework to learn a posterior distribution given the user feedback, by looking at the mutual information. Lastly, this method is incorporated in a graph-based recourse method, that is evaluated on real-world datasets.

**Strengths:**

The paper explains the problem clearly and concisely, motivating the various methodological choices. It is also quite understandable and grounded in theory. The problem itself is very interesting and I think it has significant implications for the algorithmic recourse literature.

**Weaknesses:**

I think this paper fails to discuss and compare some important baselines, and this diminishes the novelty and soundness of the approach. [1] tries to give personalized recourse options by learning a cost function from experts, which would provide a strong baseline to begin with. Secondly, [2] provides a Bayesian framework for learning personalized cost functions for algorithmic recourse, via pairwise queries, robust to user uncertainty.

More specifically, it is not clear to me the advantage or the originality of the proposed Bayesian PR over the method in [2], which is cited by the authors in the paper, but not discussed anywhere in the main manuscript.

The same concerns apply to the experimental section. I believe [1] and [2] are closely related baselines which the authors should consider testing against.

[1] Rawal & Lakkaraju. "Beyond individualized recourse: Interpretable and interactive summaries of actionable recourses." NeurIPS (2020)

[2] De Toni et al. “Personalized Algorithmic Recourse with Preference Elicitation”, arXiv preprint arXiv:2205.13743 (2022)

**Questions:**

- What is the difference between the proposed Bayesian PR and the method proposed in [2]? After a closer look, they are closely related and I would appreciate hearing how the proposed approach differs from the one presented in [2].

---

> ### Author Response · Authors · 2023-11-22
> **Response to Reviewer yvPj**
>
> Thank you for taking the time to review our work. Your questions will be thoroughly addressed in the following responses.
>
> ---
> **Q1:** I think this paper fails to discuss and compare some important baselines, and this diminishes the novelty and soundness of the approach. \
> **Response:** Thank you for highlighting the need for more extensive baseline comparisons. We acknowledge that the AReS model from [1], which focuses on global counterfactual explanations for wide-ranging population segments, offers a different perspective with its interpretable recourse summaries. Recognizing its relevance, we plan to include this model in our related work section and use it as a baseline in our experiments. This approach will help in generating a more informed prior for question determination, addressing the cold-start issue effectively. Regarding the distinctions between our work and [2], we have provided a detailed explanation in our response to the following Q2.
>
> ---
> **Q2:** What is the difference between the proposed Bayesian PR and the method proposed in [2]? After a closer look, they are closely related and I would appreciate hearing how the proposed approach differs from the one presented in [2]. \
> **Response:** Thank you for your inquiry. We have summarized the difference between our work and [2] as follows:
>
> * **Difference in personalized cost function:** Unlike [2], which utilizes a linear Structural Causal Model with parameterized weight for the cost function, we employ the Mahalanobis distance to model our cost function. The reasons for choosing Mahalanobis distance and its advantages over the $\ell_1$ distance are detailed in Section 2.
> * **Difference in problem determination:** The approach in [2] involves greedy optimization to identify an informative choice set that maximizes the Expected Utility Score (EUS), akin to listwise comparison. In contrast, our paper primarily addresses pairwise comparison, with an extension to listwise comparison discussed in Appendix A. Our goal is to identify item pairs that maximize the mutual information with the preference matrix, considering its prior distribution. We also discuss the computational benefits of pairwise over listwise comparison in Appendix A.
> * **Difference in posterior update:** Contrary to [2]'s Bayesian inference method for inferring the posterior over weights, our method updates the posterior over the preference matrix by balancing prior-posterior distortion minimization and likelihood maximization. This approach enhances the robustness of our framework, especially in the face of user response inconsistencies.
> * **Difference in recourse generation:** Our recourse generation method is graph-based, following the FACE method, as opposed to [2]'s reliance on a reinforcement learning-based approach, W-FARE.
>
> The outlined distinctions highlight the unique contributions and innovative aspects of our Bayesian PR method in contrast to the methodology employed in [2]. To provide a more thorough understanding, we will incorporate this comparative analysis into both the discussion of related work and the experimental sections of our paper.
>
> We conduct a performance comparison involving Bayesian PR, PEAR [2], and FACE on the Adult dataset. For each method, we generate the user's cost matrix $A_0$ as detailed in Section 5. Subsequently, we report the average cost and deviation across 10 users for two scenarios: using the Mahalanobis distance and the $\ell_1$ norm as the true cost functions, respectively. Notably, our Bayesian PR method achieves the lowest cost among the evaluated methods.
>
> Table: Comparison of Bayesian PR, PEAR, and FACE using Mahalanobis distance and $\ell_1$ norm as the cost
>
> | Method | Mahalanobis distance | $\ell_1$ norm |
> | --- | --- | --- |
> | Bayesian PR | **1.16 $\pm$ 0.67** | **1.45 $\pm$ 1.07** |
> | PEAR | 1.22 $\pm$ 0.59 | 1.84 $\pm$ 1.25 |
> | FACE | 1.56 $\pm$ 0.29 | 1.68 $\pm$ 0.54 |

---

### Official Review · Reviewer_tJ3D · 2023-10-30

**Soundness:** 2 fair
**Presentation:** 3 good
**Contribution:** 2 fair
**Rating:** 5
**Confidence:** 4

**Summary:**

In recourse recommendation the system suggests a sequence of feature realizations that leads from a negative classification to a positive one. Each step in the sequence is associated with personalized cost. The goal of the agent is to learn the user's cost (preferences) while simultaneously directing them to a positive state, incurring minimal overall cost on the way.
This paper proposes an approach for recourse recommendation which uses Bayesian preference elicitation for learning the user's preferences. An approximate procedure for transition selection is presented as well as an approximate posterior update. Numerical results on a few datasets are included.

**Strengths:**

* Personalized recourse recommendation is an interesting problem
* A Bayesian preference elicitation approach to personalization is plausible
* The paper is clearly written and I did not find any errors in the derivations

**Weaknesses:**

* Missing complexity analysis
* Missing comparison to sampling and evaluation of approximation quality
* Some technical issues (eg, kappa goes to inf)

**Questions:**

* Theorem 3.2 assumes that kappa goes to infinity which corresponds to noiseless responses.
  * How to understand Eq (3) (and the rest of section 4) with kappa=inf?
  * Also, the link function is approximated by a linear function (section 4.1), so how would that work when kappa=inf?
  * May be good to show empirically how this assumption affects the results vs using final kappa.
* There are several layers of approximation (eg, kappa=inf, approximating sigmoid with a linear function, finite iterations for the projection in Alg 1), it would be helpful to clearly state all of them in a single place before giving all the details. It may also be helpful to do some ablation studies and see how each level of approximation affects the overall performance.
* How does sampling compare to the analytical solution?
  * A comparison to sampling is missing from the experiments. This is important for validating the quality of the approximate analytical solution as well as the computational efficiency claims.
* Complexity:
  * It would be good to state the overall complexity of: pair choice, posterior update, and recourse generation
  * Solving an optimization problem for each possible value of m (|m_{t-1}-d| values) seems expensive
  * Comparing N^2 pairs each turn seems expensive
  * Problem 7b is an integer linear program, which is NP-hard in general, so the statement about the solution being computationally efficient seem exaggerated (especially with no detailed complexity analysis). It should be at least possible to show empirically which problem sizes can be handled and at what cost.
* Choice of x points:
  * Does it make sense to consider negative-negative pairs? Or choosing unlabeled points for comparison? It would be good to justify the positive-positive choice.
  * The graph-based approach of section 5  considers only transitions between points from the training data, but it may be possible to transition to new points not in the data (possibly unrealizable). Can such transitions be accommodated in your framework? If we have access to the classifier we can know the label of any point.
  * What are typical path lengths in the experiments?
* Section 6.1:
  * What is the dimension d?
  * The model seems rather small (90 neurons in 3 MLP layers)
  * What is the justification for the choice \tau\kappa=1?

Minor/Typos:
==========
* Eq (2): I could not find b
* Eq (3): nit - missing “(“ for R_ij
* Why not define z_i and z_j earlier and simplify notation, for example in \sigma_i and \sigma_j, and in M_ij?
* Showing validity results in Tables 1 and 2 (and 4 and 5) seems redundant since they are always 1. This can be stated once in the text. Also, x_r is chosen such that it returns a positive prediction, so it is not clear when that would not be the case.
* Section 6.3: “consistent decrease as the number of questions increases” is a bit overstated, it seems that in most plots the rank increases initially before decreasing.

**Details Of Ethics Concerns:**

There is a significant overlap with submission 8776.

---

> ### Author Response · Authors · 2023-11-22
> **Response to Reviewer tJ3D [1/3]**
>
> Your review and feedback are highly valued, and we will be addressing your questions in the subsequent sections.
>
> ---
> **Q1:**  How to understand Eq (3) (and the rest of section 4) with kappa=inf? Also, the link function is approximated by a linear function (section 4.1), so how would that work when kappa=inf?\
> **Response:** Thank you for your question. When considering $\kappa=\infty$, as outlined in the proof of Proposition 1 and by applying the Dominated Convergence Theorem, we derive the following limit:
> \begin{equation*}
> \lim_ {\kappa\rightarrow \infty} \text{Prob}(\tilde R_{ij}^\kappa(\tilde A)\neq R_{ij}) = \int_ {S_+^d} \lim_ {\kappa \rightarrow \infty} \Phi(\kappa R_{ij}\Delta_{ij}(S))f_P(S) dS = \int_ {S_+^d} \mathrm{1}_ {\{R_{ij}\Delta_{ij}(S)\geq 0\}} f_P(S)dS.
> \end{equation*}
> In this scenario, we can also use the linear function $v \mapsto v$ to approximate the step function $v \mapsto \mathrm{1}_ {\{v \geq 0\}}$. This approximation leads us to the following expression when $\kappa = \infty$
> \begin{equation*}
> \text{Prob}(\tilde R_{ij}^\kappa(\tilde A)\neq R_{ij}) \approx E_P[R_{ij}\langle M_{ij}, \tilde A\rangle].
> \end{equation*}
> Accordingly, we can set $\tau=1$ in this context. We will include the discussion of the case $\kappa=\infty$ in the Appendix.
>
> ---
> **Q2:**  May be good to show empirically how this assumption affects the results vs using final kappa. \
> **Response:** We have empirically tested the impact of the $\kappa$ assumption, showcased in Figures 4 to 7 in Appendix D. These histograms display Mahalanobis distances for different $\kappa$ values ($+\infty$, 1, 5, 10) at $T=10$. The results reveal a consistent trend: with increasing $T$, the distribution means converge towards the true cost values for all $\kappa$ values, accompanied by reduced standard deviations. This indicates our framework's robustness and precision, effectively handling variations in finite $\kappa$.
>
> ---
> **Q3:** There are several layers of approximation, it would be helpful to clearly state all of them in a single place before giving all the details. It may also be helpful to do some ablation studies and see how each level of approximation affects the overall performance. \
> **Response:** Thank you for the suggestion. Our method primarily involves two approximations: first, we consider the case $\kappa = \infty$ in mutual information computation for an analytical expression (Section 3); second, we approximate the sigmoid function with a linear function for posterior updates. The computational error in Algorithm 1, controlled by the stopping criteria $\\|\Sigma_{(k)}-\Sigma_{(k-1)}\\|<10^{-6}$, is minimal. Empirical results (Figures 4 to 7) show that as iterations increase, the approximations yield values closer to the true costs for all datasets and $\kappa$ values, with diminishing standard deviations. Detailed ablation studies for each approximation level remain a potential future research direction.

---

> ### Author Response · Authors · 2023-11-22
> **Response to Reviewer tJ3D [2/3]**
>
> **Q4:** How does sampling compare to the analytical solution? A comparison to sampling is missing from the experiments. This is important for validating the quality of the approximate analytical solution as well as the computational efficiency claims. \
> **Response:** Thank you for highlighting this aspect. Our discussion following Theorem 3.2 outlines the computational efficiency of the analytical approach over sampling, particularly in terms of time complexity. Regarding the validation of the approximation's quality, we have presented histograms in the experimental section that illustrate Mahalanobis distances for various $\kappa$ values ($+\infty$, 1, 5, 10) at $T=10$. These findings indicate a clear trend: as $T$ increases, the means of these distributions consistently align closer to the actual cost values for all $\kappa$ values, with a corresponding reduction in standard deviations. Acknowledging the importance of your suggestion, we incorporate a comparison with sampling in future revisions, focusing on both the quality of the results and execution time.
>
> We test the estimated probability and the analytic probability on four different dimensionalities. For each dimensionality, we perform Monte Carlo sampling with 100000 samples and evaluate $\langle A_i, M \rangle \leq 0, ~ i = 1, \cdots, 100000$. We report the computation time for both approaches. As we can see in the following table, the estimated probabilities match the analytical probability which concludes the validity of the analytic form. Furthermore, the computation times of the estimated probabilities grow more significantly than the analytic method as we deal with higher dimensions, justifying the need to adopt analytic probability.
>
> Table: Comparison with sampling method for time and probability value
> | dimensionality | estimated-probability | estimated-time | analytic-probability | analytic-time |
> | --- | --- | --- | --- | --- |
> | 2   | 0.0020 | 1.3270 sec | 0.0019 | 0.0009 sec |
> | 5   | 0.0415 | 1.5225 sec | 0.0417 | 0.0022 sec |
> | 10  | 0.0099 | 2.7120 sec | 0.0098 | 0.0436 sec |
> | 15  | 0.9915 | 3.6238 sec | 0.9915 | 0.0516 sec |
>
> ---
> **Q5:** It would be good to state the overall complexity of pair choice, posterior update, and recourse generation. \
> **Response:** Thanks for your suggestion. Please refer to the response to Q1 in the global response part.
>
> ---
> **Q6:**  Does it make sense to consider negative-negative pairs? Or choosing unlabeled points for comparison? It would be good to justify the positive-positive choice. \
> **Response:** Our focus on positive-positive pairs is driven by the aim to alter the negative outcome to a positive outcome in prefactual recommendations; hence it is reasonable to consider only positive samples so that we can guide the user to improve. While negative-negative pairs or unlabeled points could provide additional data for the preference matrix $A$, our priority is efficiency and user experience. Our approach minimizes user queries, demonstrated in our experiments where only 10 questions are asked, thus reducing user burden.
>
> ---
> **Q7:** The graph-based approach of section 5 considers only transitions between points from the training data, but it may be possible to transition to new points not in the data (possibly unrealizable). Can such transitions be accommodated in your framework? If we have access to the classifier we can know the label of any point. \
> **Response:** Yes. Our framework can theoretically accommodate transitions to new data points not present in the training data. This would involve enumerating these new points and integrating them alongside the training data within our graph structure. However, this approach raises concerns about the actionability of such transitions, as these new data points might be unrealizable in practical scenarios. In line with maintaining actionable recourse strategies, we adhere to the FACE methodology, which constructs the graph using only training data. This decision ensures that the generated recourses are both realistic and feasible.
>
> ---
> **Q8:** What are typical path lengths in the experiments? \
> **Response:** In our experiments, the average path lengths observed for each dataset were as follows: Synthesis: 2.07, German: 2.54, Bank: 5.47, Student: 5.31.
>
> ---
> **Q9:** What is the dimension $d$? \
> **Response:** The dimension $d$ represents the dimension of $x$.
>
> ---
> **Q10:** The model seems rather small (90 neurons in 3 MLP layers).\
> **Response:**  For the classifier, we adhere to the setup used by Wachter et al. (2017). The classifier's accuracy details are included in Appendix D of our updated paper.

---

> ### Author Response · Authors · 2023-11-22
> **Response to Reviewer tJ3D [3/3]**
>
> **Q11:** What is the justification for the choice $\tau\kappa = 1$?\
> **Response:** Regarding the choice of $\tau\kappa = 1$, this is based on the inverse relationship between $\tau$ and $\kappa$: by setting $\tau$ as $1/\kappa$, we ensure their product equals 1. While alternative values for $\tau$ could be determined through cross-validation, this process can be time-intensive. Importantly, our empirical findings demonstrate that the choice of $\tau \kappa=1$ yields robust performance. Specifically, this parameter setting results in lower costs when compared to other baselines, thereby validating its efficacy within the framework of our model.
>
> ---
> **Other Comments:**\
> Thank you for the comments. We have incorporated the comments into our updated paper.

---

> ### Author Response · Authors · 2023-11-22
> **Distinction between Submission 8776 and our paper**
>
> Thank you for your comments. We appreciate the opportunity to clarify the distinctions between our work and Submission 8776. After conducting a plagiarism check with Copyleaks, we confirmed a similarity of only 5.2\%, mainly due to general headings and shared references. We have summarized the difference between our paper and submission 8776 as follows:
>
> * **Major Methodological Difference:** We utilize a Bayesian preference elicitation method to estimate the preference matrix $A$. This approach is based on probabilistic principles, where user preferences are iteratively updated and refined through Bayesian inference. It is designed to capture the variability and uncertainty inherent in individual preferences. On the contrary, Submission 8776 employs a graphic-like approach, focusing on finding a hyperplane closest to the Chebyshev center for segmenting the confidence set of $A$. This technique emphasizes geometric and spatial analysis, providing a different perspective on preference estimation.
>
> * **Handling Randomness and Inconsistency:** In our approach, we incorporate the Bradley-Terry-Luce (BTL) model to address randomness and inconsistency in responses. The BTL model offers a probabilistic way of understanding user preferences, especially in scenarios with inherent variability. Alternatively, Submission 8776 introduces a positive margin to account for potential errors in subject judgment. This method sets a fixed error threshold, offering a distinct way of managing response uncertainties.
> \end{enumerate}
>
> These differences highlight our unique approach in theory and technique compared to Submission 8776, despite targeting similar objectives in preference matrix estimation.

---

### Official Review · Reviewer_iStc · 2023-10-30

**Soundness:** 4 excellent
**Presentation:** 3 good
**Contribution:** 3 good
**Rating:** 6
**Confidence:** 4

**Summary:**

An approach for personalized algorithmic recourse based on Bayesian
preference elicitation. The apprach selects queries to the user that
maximize mutual information between the response and the cost
function, and provides an efficient solution to choose queries and
compute the posterior of the cost distribution.

**Strengths:**

A solid work presenting a methodologically sound and novel solution to
bayesian preference elicitation of cost functions, complemented with
an efficient implementation.

The manuscript is complemented with an extensive supplementary section
detailing proofs, extensions from pairwise to listwise comparisons,
and additional experimental results.

**Weaknesses:**

From an algorithmic recourse perspective, the approach builds on the
FACE algorithm, that achieves recourse by selecting a
recourse-achieving instance from the training set. This can be badly
suboptimal for users that are not similar enough to any specific
training user. While experimental results in the appendix indicate
competitive results with some alternatives using different strategies,
this limitation should be mentioned in the main paper. As far as I
understand, this limitation is not due to the proposed Bayesian
approach, but rather to the recourse strategy employed, so that
adapting the approach to deal with other recourse strategies could
further strenghten the contribution.

There is a recent research line in the algorithmic recourse community
that advocates the need for a causal perspective in algorithmic
recourse, where the effect on a action on causally dependent features
of the user state is accounted for.  Karimi et al. "(Algorithmic
Recourse: from Counterfactual Explanations to Interventions", 2020)
demonstrated that optimal algorithmic recourse cannot be achieved if
this causal perspective is ignored. This aspect is ignored in the
manuscript, making the overall approach (which is perfectly fine for
generic personalized recommendations) less appealing in the
algorithmic recourse context.

**Questions:**

Can the approach be adapted to work with recourse strategies that generalize beyond training examples?


Can you comment on the impact of the lack of a causal perspective on recourse achievement and cost minimization?

---

> ### Author Response · Authors · 2023-11-22
> **Response to Reviewer iStc**
>
> Thank you for your review and feedback. In the upcoming sections, we will address the questions you have raised.
>
> ---
> **Q1:** Can the approach be adapted to work with recourse strategies that generalize beyond training examples?\
> **Response:**  Yes.  After estimating the preference matrix $A$ with a Wishart distribution $P_T\sim \mathcal{W}(m_T,\Sigma_T)$ using the Bayesian preference elicitation framework from sections 3 and 4, we can integrate the gradient-based recourse generation method proposed by Wachter et al. (2017). In applying the gradient-based recourse generation method, we work with a given machine learning model $f_w(x)$ and a loss function $g$ to measure the difference between $f_w(x)$ and the target prediction, which is 1 in our case. The optimization problem, adapted from Wachter et al. (2017), is defined as:
> \begin{equation*}
>       \min_{x}\max_{\lambda}~\lambda g(f_w(x),1) + (x-x_0)^\top (m_T\Sigma_T)(x-x_0).
> \end{equation*}
> Here, $m_T\Sigma_T$ serves as an approximation for the preference matrix $A$. The maximization over $\lambda$ is achieved by iteratively solving for $x$ and increasing $\lambda$ to find a close solution, as described in Wachter et al. (2017). We can also explore other methods like gradient descent-ascent in minimax problems or considering the dual problem. Exploring the extension of our framework to encompass scenarios beyond the training samples, along with these alternative solution methods, could be a potential direction for our future research.
>
> ---
> **Q2:** Can you comment on the impact of the lack of a causal perspective on recourse achievement and cost minimization?\
> **Response:** We acknowledge the theoretical importance of the causal perspective on the recourse problem as highlighted by Karimi et al., which we refer to in our introduction. However, our approach pragmatically navigates the complexities and variability of expert opinions, as well as the challenges in scaling this method to larger networks. This decision reflects the practical difficulties in establishing a universally accepted causal graph, especially in contexts with diverse expert perspectives or in expansive networks. Additionally, certain datasets, like the adult dataset, often lack a clear, consensus-driven causal graph, as discussed by Mahajan et al. [A]. This further underscores the adaptability of our methodology in scenarios where a causal approach may not be feasible or effective.
>
> [A] Mahajan, Divyat, Chenhao Tan, and Amit Sharma. ''Preserving causal constraints in counterfactual explanations for machine learning classifiers.'' arXiv preprint arXiv:1912.03277 (2019).

---

### Official Review · Reviewer_mLqk · 2023-11-04

**Soundness:** 2 fair
**Presentation:** 1 poor
**Contribution:** 2 fair
**Rating:** 3
**Confidence:** 3

**Summary:**

This paper proposes the so-called Prefactual Recommendation by Bayesian Preference Elicitation algorithm for personalized recommendation.

**Strengths:**

Personalized recommendation is an important question in the field of learning to rank.

**Weaknesses:**

1. Presentation is poor (see my questions below)
2. The paper focuses on binary class only, which constraints the application of this algorithm
3. No significant improvement in comparison with the baseline.
4. The theoretical analysis may not be enough (see my questions below)

**Questions:**

------------------------- For Presentation -------------------------

1. It's not clear enough why this paper applies Mahalanobis distance is selected. Why does this distance make sense to serve as the cost function? How is it compared with many other distance functions and other cost functions?

2. The main message of the main theorem of this paper is hard to follow. In (2) of Theorem 3.2, I don't believe any author could easily figure out the asymptotic manner of this probability (either it's a constant, close to 0, or close to 1?).

3. Following 2, the complexity of the proposed algorithm is unclear and there is no analysis on how many samples (how large $M$) are needed for the algorithm.

4. It's not clear what is the difference between the proposed algorithm with FACE. Does FACE also use shortest path? If so, what's their cost?

------------------------- For Experiments -------------------------

1. I don't think the proposed algorithm is better than the baselines because it has almost the same cost as FACE if using $\ell_1$ as the true cost. The improvement in Mahalanobis distance as the true cost is not compatible because this is the same distance used in the algorithm and to some extent, the algorithm cracks the ground truth.

 2. Why the baselines are not compared in Section 6.3?

3. (Minor) I suggest moving other baselines (Appendix D.2) to the main text.


------------------------- For Theoretical Analysis -------------------------

1. The complexity analysis of the whole proposed algorithm is missing or lacks discussion.

2. (Minor) Asymptotic mutual information is missing citation.

---

> ### Author Response · Authors · 2023-11-22
> **Response to Reviewer mLqk [1/2]**
>
> Your review is appreciated, and we would like to express our gratitude. We will proceed to address your questions in the forthcoming sections.
>
> ---
> **For Presentation:**\
> **Q1:** It's not clear enough why this paper applies Mahalanobis distance is selected. Why does this distance make sense to serve as the cost function? How is it compared with many other distance functions and other cost functions? \
> **Response:** In Section 2, we justify our choice of Mahalanobis distance over Euclidean or Manhattan ($\ell_1$) distances for preference modeling. This distance stands out for its ability to account for variances and correlations among dimensions, providing a nuanced view of individual, non-uniform preferences and subjective 'closeness', which Euclidean distance, treating all dimensions equally, fails to capture. Additionally, Mahalanobis distance incorporates a personalized matrix $A$, relevant in scenarios where costs involve cognitive elements and complex preference structures. This metric's effectiveness is widely recognized in the field of metric learning, as evidenced by literature like Kulis (2013), Yang \& Jin (2006), and Bellet et al. (2013). Its adaptability to unique data distributions makes it particularly suitable for intricate, subjective decision-making modeling.
>
> Furthermore, minimizing this distance aligns with the innate human tendency to achieve desired outcomes with the least effort. By using the Mahalanobis distance, we create a Riemannian manifold, introducing an element of curvature through the preference matrix $A$ [A]. This approach effectively encapsulates the complex interplay between preference and cost, enabling us to accurately model the diverse aspects of individual decision-making.
>
>
> [A] Arvanitidis, Georgios, Lars K. Hansen, and S$ø$ren Hauberg. "A locally adaptive normal distribution." Advances in Neural Information Processing Systems 29 (2016).
>
> **Q2:** The main message of the main theorem of this paper is hard to follow. In (2) of Theorem 3.2, I don't believe any author could easily figure out the asymptotic manner of this probability (either it's a constant, close to 0, or close to 1?).\
> **Response:** We apologize for any confusion caused. The main intent of Theorem 3.2 is to establish an analytical formula for calculating the probability $\mathbb{P}(\Delta_{ij}(\tilde A))$. This probability is a constant value when the items $x_i$ and $x_j$ and the parameters of the Wishart distribution, namely the degrees of freedom $m_{t-1}$ and scale matrix $\Sigma_{t-1}$ are given.
>
> **Q3:** Following 2, the complexity of the proposed algorithm is unclear and there is no analysis on how many samples (how large $M$) are needed for the algorithm. \
> **Response:** As for the complexity of the proposed algorithm, please refer to the response to Q1 in the global response section. Regarding the number of samples ($M$), our algorithm doesn't impose a specific limit. The choice of $M$ affects performance and computational efficiency: larger $M$ enhances robustness but increases computation, while smaller $M$ reduces computational load but may affect outcomes.
>
> **Q4:**  It's not clear what is the difference between the proposed algorithm with FACE. Does FACE also use the shortest path? If so, what's their cost? \
> **Response:** The primary distinction between our proposed algorithm and FACE lies in the modeling of the cost function. FACE assumes a uniform $\ell_1$ norm as the cost function for all users, which does not account for the diversity in individual cost preferences. In contrast, our approach utilizes the Mahalanobis distance to model a personalized cost function. We employ Bayesian preference elicitation to accurately estimate the weighting matrix $A$, tailoring the cost function to individual user preferences.
>
> Both our method and FACE use the shortest path approach in their respective algorithms. However, the key difference is in how the cost associated with each path is calculated and personalized, which in our model is achieved through the Mahalanobis distance and the estimated matrix $A$, as opposed to the standard $\ell_1$ norm used by FACE.

---

> ### Author Response · Authors · 2023-11-22
> **Response to Reviewer mLqk [2/2]**
>
> **For Experiments:**\
> **Q1:** I don't think the proposed algorithm is better than the baselines because it has almost the same cost as FACE if using as the true cost. The improvement in Mahalanobis distance as the true cost is not compatible because this is the same distance used in the algorithm and to some extent, the algorithm cracks the ground truth. \
> **Response:**  We apologize for any misunderstanding. In the context of Table 1, we specifically use Mahalanobis distance as the true cost. In this setup, our framework accurately specifies the cost function type to align with the Mahalanobis distance, whereas FACE misspecifies it. As a result, our method demonstrates improved cost efficiency compared to FACE when Mahalanobis distance is the true cost.
>
> Turning to Table 2, where the true cost is modeled as the $\ell_1$ norm, FACE specifies the cost function correctly, while our framework does not. Nonetheless, even with this misspecification, our method exhibits competitive performance, matching or surpassing FACE in cost efficiency on the Synthetic and Student datasets. This contrast between Tables 1 and 2 underscores the adaptability and resilience of our approach in handling various cost function scenarios.
>
> **Q2:** Why the baselines are not compared in Section 6.3? \
> **Response:** In Section 6.3, we focused on comparing our method with the graph-based FACE method due to its direct relevance. Comparisons with non-graph-based methods are detailed in the Appendix, owing to space constraints in the main text, ensuring a comprehensive yet concise presentation.
>
> ---
> **For Theoretical Analysis:**\
> **Q1:** The complexity analysis of the whole proposed algorithm is missing or lacks discussion. \
> **Response:** Please refer to the response to Q1 in the global response part.
>
> ---
> **Other Comments:**\
> Thanks for the comments. We incorporate the comments in the our updated paper.

---

### Author Response · Authors · 2023-11-22
**Global Response**

Thanks for all the insightful feedback. We summarize the common question and provide corresponding responses in this global response section. We would rather happy to take any other questions during the discussion period.

**Q1:** The complexity analysis of the whole proposed algorithm is missing or lacks discussion.

**Response:** Thank you for pointing out the need for a detailed complexity analysis.

- **Complexity of question determination:** In our discussion following Theorem 3.2, we address the complexity involved in question determination. The computational complexity for determining the analytical form of the probability is $\mathcal{O}(d^2)$, where $d$ is the dimension of the data samples. Considering the number of pairs of items, which is $N(N+1)/2$, the overall complexity of question determination is $\mathcal{O}(TN^2d^2)$, where $T$ is the number of questions asked. In our experiments, $T$ is set to 10. Our method's time complexity is less than the complexity $\mathcal{O}(TN^2Ld^2)$ for sampling methods. The difference is mainly due to $L$, the sample size, which greatly increases complexity in sampling-based approaches.

- **Complexity of posterior update:** Lemma 4.5 indicates that Algorithm 1 achieves a linear convergence rate. For each iteration, the time complexity amounts to $\mathcal{O}(d^3 + d \log(d))$. This is derived from the $\mathcal{O}(d^3)$ complexity for inverting a $d$-by-$d$ matrix and $\mathcal{O}(d\log(d))$ for computing the projection. Consequently, the complexity required to achieve an $\varepsilon$ approximation is $\mathcal{O}(d^3\log(1/\varepsilon))$. To infer the optimal integer $d \leq m^\ast \leq m_{t-1}$ such that $(m^\ast, \Sigma^\ast(m^\ast))$ minimizes the loss function $L(m, \Sigma)$, the overall complexity is $\mathcal{O}((m_{t-1}-d)d^3\log(1/\varepsilon))$.

- **Complexity of recourse recommendation:** The recourse recommendation problem in our framework is analogous to the shortest path problem. To address this, we adopt the approach used in FACE, employing Dijkstra’s algorithm for its resolution. The time complexity of Dijkstra’s algorithm is $\mathcal{O}(|E| + |V|\log |V|)$, where $|E|$ is the number of edges and $|V|$ is the number of nodes. In our specific context, considering the worst-case scenario, the time complexity can escalate to $\mathcal{O}((N+M)^2)$.

---

### Meta-Review · Area_Chair_RsQ9 · 2023-12-06

**Metareview:**

The paper proposes a method for learning personalized cost functions based on (active) queries of pairwise comparisons, and then do recourse recommendation. The proposed framework is evaluated empirically.

Pros: the problem is important. The authors tried to address the concerns in detailed responses.

Cons: Reviewers raised concerns about the lack of novelty (while some reviewers found the idea novel), insufficient theoretical analysis, lack of comparison with the literature, and marginal improvement over the baseline. Eventually, no reviewers were excited enough to champion the paper.

Concerns on ethical issues. Multiple reviewers mentioned that this paper shares many common features with Submission 8776, including the problem studied, the references, and the dataset. In particular, the following reg flags were raised.

1. The two submissions used different wordings of the same problem: PREFACTUAL RECOMMENDATION vs RECOURSE RECOMMENDATION.

2. The introductions, especially the motivation, of the two papers are similar and the main difference is in wording.

3. The mathematical definitions of the same problem look unnecessarily different.

4. The overall approaches look similar, in particular the Figure 1 in #4706 and Figure 3 in #8776 look similar, so it is hard to tell how different the ideas behind them are.

These look suspicious because using the same names and mathematical definitions of an existing technical problem is not viewed as plagiarism. We understand that the authors' worry of using similar or the same text. Still, the extent to which the two papers are different in places they should not was more than we typically see.

The reviewers thought that the situation can be handled in a better way, as one of them commented "The papers tackle the very same problem, and in principle a comparison between them should be expected if they come from the same authors". At some point a reviewer proposed that the authors should merge the two papers in the first place. The AC went through both submissions and agreed with the reviewers.

The authors responded to the ethical concern in the rebuttal and highlighted the technical differences, which was helpful. After the rebuttal and discussions, the reviewers believe that the proposed technical approaches are sufficiently different. One thing to note is that during the rebuttal (of #8776), the authors mentioned that Copyleaks gives a low score for similarity. This was not viewed as convincing, especially given that the two papers are unnecessarily different in many aspects. In fact, the AC felt that using such evidence could potentially make the paper appear more suspicious, as it could give people an impression that the papers were written to circumvent plagiarism checkers. Eventually, the recommendation was mostly based on the perceived qualities of the two submissions.

That being said, after discussions among the reviewers and among AC, SAC, and chairs, we agreed that this discussion should be conveyed to the authors, and the authors are advised to be more careful about similar issues in future submissions. As one reviewer mentioned, perhaps the authors can consider merging the two submissions to create a stronger and more thorough paper in the future.

We hope this is helpful, and thanks again for submitting to ICLR!

**Justification For Why Not Higher Score:**

The Cons in the metareview.

**Justification For Why Not Lower Score:**

N/A

---

### Decision · Program_Chairs · 2024-01-16

Reject